# MECHANISTIC INSIGHTS: CIRCUIT TRANSFORMATIONS ACROSS INPUT AND FINE-TUNING LANDSCAPES

## ABSTRACT

Mechanistic interpretability seeks to uncover the internal mechanisms of Large Language Models (LLMs) by identifying circuits—subgraphs in the model's computational graph that correspond to specific behaviors—while ensuring sparsity and maintaining task performance. Although automated methods have made massive circuit discovery feasible, determining the functionalities of circuit components still requires manual effort, limiting scalability and efficiency. To address this, we propose a novel framework that accelerates circuit discovery and analysis. Building on methods like edge pruning, our framework introduces circuit selection, comparison, attention grouping, and logit clustering to investigate the intended functionalities of circuit components. By focusing on what components aim to achieve, rather than their direct causal effects, this framework streamlines the process of understanding interpretability, reduces manual labor, and scales the analysis of model behaviors across various tasks. Inspired by observing circuit variations when models are fine-tuned or prompts are tweaked (while maintaining the same task type), we apply our framework to explore these variations across four PEFT methods and full fine-tuning on two well-known tasks. Our results suggest that while fine-tuning generally preserves the structure of the mechanism for solving tasks, individual circuit components may not retain their original intended functionalities.

## 1 INTRODUCTION

The rapid advancement of large language models (LLMs) has established them as powerful tools across a wide range of tasks for (Vaswani, 2017; Devlin, 2018; Achiam et al., 2023) natural language and beyond. However, their vast scale—often involving billions of parameters—and black-box nature pose significant challenges in understanding their decision-making processes, leading to unpredictable risks in critical domains where accuracy, fairness, and transparency are essential. The field of interpretability aims to address these challenges by developing methods to uncover the internal reasoning of these models. Mechanistic interpretability, a subfield within this area, seeks to reverse-engineer LLMs into human-understandable algorithms (Conmy et al., 2023; Bereska & Gavves, 2024). A key objective is to identify **circuits**, subgraphs within the model's computational graph that correspond to specific behaviors, while maintaining a high level of sparsity and preserving the model's performance on the given task.

Recent progress in mechanistic interpretability has shed light on the inner workings of language models through meticulous human inspections, uncovering key circuits responsible for specific tasks (Wang et al., 2022; Hanna et al., 2024; Merullo et al., 2023). To accelerate this process and reduce reliance on costly manual efforts, automated tools have been developed to systematically identify circuits driving certain behaviors (Conmy et al., 2023; Bhaskar et al., 2024; Syed et al., 2023). However, these methods are either time-consuming (Conmy et al., 2023) or rely on approximations that prioritize speed over reliability (Syed et al., 2023). As a result, further investigations into the relationships between circuits (Tigges et al., 2024; Prakash et al., 2024; Jain et al., 2024) are limited in either scale or precision in finding circuits. The recent introduction of techniques like Edge Pruning (Bhaskar et al., 2024) has made it possible to automatically identify more accurate and faithful circuits across larger datasets while keeping computational costs manageable.

Despite advancements in circuit discovery, significant challenges remain in understanding the functionalities of the circuit components. Existing studies often assign specific functions to individual nodes within circuits through manual inspection. These nodes and their roles are typically identified during the discovery process by observing changes in the model's output (e.g., variations in logit values) when specific node activations are perturbed. This approach generally requires extensive path interventions to determine the direct effects of nodes on the final output and careful design of such intervention. The process demands considerable manual effort, leading to a limited scope (e.g. only focus on the change in target output logits) and placing a heavy burden on researchers, ultimately slowing down the progress of Mechanistic Interpretability. Moreover, as circuit discovery methods become automated, the identification of circuits and their functionalities no longer happen in parallel. While progress has been made in circuit identification, a growing gap persists in understanding the broader implications —*after circuits are identified, what coming next?* With automated methodologies generating vast amounts of circuits, investigating their internal functionalities has become increasingly laborious and challenging.

This challenge is evident in our observations from the IOI task. Modifying objects within the task results in significant changes to the discovered circuits, as shown in Fig. 1. Intuitively, circuits responsible for the same reasoning across similar tasks should remain consistent. However, the pretrained model struggles to maintain this consistency when performing identical tasks across various types of objects (e.g., changing "Mary" to "Dog"). Moreover, applying different supervised finetuning methods introduces further variations in the circuits. However, investigating these changes in circuits and their functionalities becomes increasingly demanding and labor-intensive.

Inspired by these challenges and building on these findings, this work explores how circuits vary in both structure and functionality across different ablated prompts, and how model fine-tuning methods affect these circuits. To address this, we propose a framework for circuit analysis that integrates **efficient discovery** with **automated interpretability**. To broaden the scope of functionality exploration and accelerate the process, we make a trade-off by foregoing functionality conclusions based on causal effects. Instead, we integrated the attention pattern and logit lens with clustering techniques to describe the **intended functionalities** of the circuits components. By 'intended functionalities', we refer to what the circuit components are attempting to do in the context of a given tasks, rather than what they ultimately contribute to the final output. Our framework extends the edge pruning algorithm (Bhaskar et al., 2024) for circuit discovery by incorporating stages for **circuit selection**, **circuit comparison**, **attention grouping**, and **logit clustering**, as outlined in Fig. 2. We further summarize our findings and contributions as following: (1). We introduce a novel framework that accelerates the process of estimating the functionality of circuit components, reducing the reliance on manual efforts. (2). We demonstrate that the functionality of circuit components does not necessarily remain consistent across ablated prompts. (3). We show that while fine-tuning methods maintain the overall functional structure, the specific components performing those functions may change.

## 2 EXPERIMENT SETUP AND PIPELINE

In this section, we first outline the experimental setup, providing an overview of the prompt settings for the tasks and fine-tuning methodologies involved in our work. Next, we describe the process for identifying and selecting circuits, along with the metrics used for circuit validations. We then introduce the methods for evaluating the intended functionality of the circuit components. Lastly, we discuss how comparative circuit analysis is conducted, following the pipeline in Fig. 2.

### 2.1 TASKS AND MODELS

We explore circuit variations within the context of two specific tasks: Indirect Object Identification (IOI) and Greater Than (GT). These tasks were initially examined by Wang et al. and Hanna et al., respectively. We focused on IOI and GT due to their well-established, human-inspected circuits, making them particularly suitable for implementing our framework and conducting an in-depth analysis of circuit variations through different fine-tuning methods and ablated prompts. Additionally, recent work by Merullo et al. offers promising insights through the study of these fundamental tasks. Building on top of these tasks, Merullo et al. has studied the transferrability of these circuits beyond the base syntactic structure of these tasks. To gain a more granular understanding of the

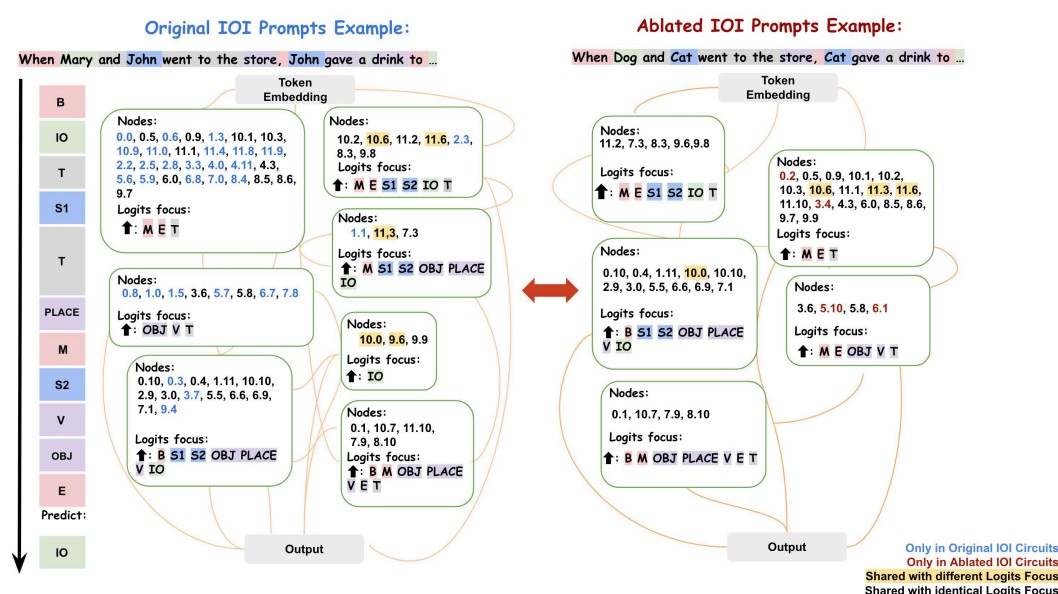

Figure 1: We demonstrate how circuits vary between IOI prompts and ablated prompts. When human names are replaced with animals, the newly discovered circuits become almost a subset of the IOI circuits. Nodes are clustered based on their logit outputs. We focus on logit outputs regarding to the prompt components. For each cluster, the major up-weights on logits for the sentence components are listed. Upon further analysis, we found that even overlapping nodes can perform different functions, colored in yellow.

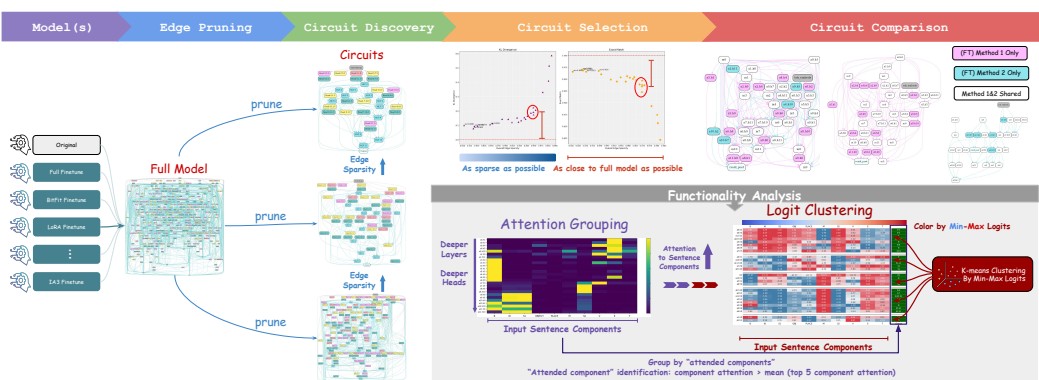

Figure 2: Pipeline of experiments: (1) Fine-tuned models on certain tasks; control model being the original GPT-2 model (2) Apply Edge-pruning to identify a series of candidate circuits (3) Select ideal circuits that balance sparsity and performance recovery, evaluated using KL Divergence and Exact Match metrics. (4) Compare circuits across models fine-tuned with different methods. (5) Conduct functionality analysis by: (5a). Grouping the heads based on their primary attended input sentence components - what the head is focusing on. (5b). Computing logits for each input sentence component per head - which components the head up-weights or down-weights for generation. (5c). Clustering heads by their min-maxed logits across input sentence components - which heads share or differ in functionality.

model's mechanisms, we focus on GPT-2 small, as its resulted circuits are more manageable and human-readable. More details are discussed as follows:

**Indirect Object Identification (IOI):** IOI (Wang et al., 2022) is the task of predicting the name of the indirect object in a sentence. The task follows the format of "When {A} *and* {B} *went to* {PLACE}, {B} *bought a* {OBJECT} *to* → {A}". The datasets are generated by filling in A, B, PLACE, and OBJECT with a list of nouns. The model then completes the sentence by filling in the predicted indirect object, ideally {A}. Additionally, the prompts include variations in both ABBA and BABA formats. In the original IOI settings, A and B are filled by human names such as "John" and "Mary" as shown in Fig. 1. Following the setting by Edge Pruning (Bhaskar et al., 2024), we adopt the prompts on a variant with 30 templates from Hugging Face. Similarly, we randomly select 200 examples each for the train and validations; and 36,084 instances for the test sets.

**Ablated IOI:** In addition to the IOI dataset, we introduce an ablated IOI dataset to investigate whether the functionality and structure within the circuits are preserved across different types of objects, inspired by the observations in Fig. 1. We retain the template structure from the original IOI tasks; however, instead of populating A and B with human names, we use capitalized names of animals, cities, and colors. An example of the ablated prompt can be found in Fig. 1. To avoid tokenization issues that could affect model performance, we only include names that map to a single token. All other settings, such as train, test, and validation splits, remain the same as in the original IOI tasks.

**Greater Than (GT):** GT tasks (Hanna et al., 2024) follows the format of "*The war lasted from the year 1743 to 17* → *xy*." The tasks requires the model to place a higher probability on the continuations 44, 45, ..., 99, compared to 00, 01, ..., 42. We adopted the version of dataset proposed in Edge Pruning (Bhaskar et al., 2024) which has 150 examples in the train and valitation, and 12,240 instances in the test. This task is generally considered simpler than IOI, as it involves fewer logical steps to complete.

**PEFT Methods:** We primarily focus on Parameter-Efficient Fine-Tuning (PEFT) methods that preserve the original model structure. Specifically, we fine-tuned the GPT-2 small model with supervision, using Bitfit (Zaken et al., 2021), LoRA (Hu et al., 2021), IA3 (Liu et al., 2022), and AdaLoRA (Zhang et al., 2023), on the IOI and GT tasks. A fully fine-tuned model was used as a control for comparison. To ensure a fair comparison, all fine-tuned models are controlled to achieve nearly identical performances on the same test set. For the IOI task, the goal was to minimize the prediction loss for the indirect object. For the GT task, the goal was to maximize the gap between the cumulative probabilities of correct and incorrect years.

## 2.2 CIRCUIT DISCOVERY AND SELECTION

We adopted edge-pruning (Bhaskar et al., 2024) to automate the circuits discovery step for each finetuned models along with the pretrained version across the above mentioned tasks. To evaluate and ensure the faithfulness and preciseness of the identified circuits, we mainly relied on the measure of KL-Divergence and Exact Match, adopted by Bhaskar et al. and Conmy et al..

**Circuit Discovery Algorithm:** We implemented the edge pruning algorithm for automated circuit discovery (Bhaskar et al., 2024). This method addresses circuit discovery through gradient-based pruning on the edges of the model's computational graph over hyperparameter *edge-sparsity (es)*. Similar to path-patching (Wang et al., 2022) and the other automated methodologies (Conmy et al., 2023; Syed et al., 2023), it involves causal interventions on edges by substituting the target edges with counterfactual activations from corrupted examples. This process further generates sparse circuits by masking out edges that has no causal effect on the target tasks.

In this work, we first reproduced the circuits for the IOI and GT tasks. Next, we retrieved the circuits from the models fine-tuned with supervision on the two tasks. Finally, we retrieved the circuits from the models on the ablated IOI dataset using all the described models.

**Circuit Evaluation and Selection:** A circuit is considered accurate and faithful when its output closely aligns with that of the full model, even at a high level of graph sparsity, as highlighted in previous works (Hanna et al., 2024; Conmy et al., 2023; Bhaskar et al., 2024). Specifically, we utilized metrics, such as Exact Match for IOI, Kendall's $\tau$ for GT, and KL Divergence for both, from the edge pruning approach to assess how closely the circuit's output aligns with the full model. Addi-

tionally, we measured the differences between target and distractor outputs, such as logit difference for IOI and probability difference for GT, both traditionally used for these tasks.

As shown in the circuit selection stage of Fig. 2, we observed an exponential relationship between edge sparsity and KL divergence, consistent across all models and tasks. More results for the metrics are provided in the Appendix B. For this work, we selected a group of 'best' circuits for each model by identifying those at the 'knee' of the KL-divergence curve, as highlighted in red in the circuit selection stage of Fig. 2. Using this selection criterion, we chose the circuits with the highest possible sparsity, just before the KL divergence sharply increases. Among the circuits in this selected group, we considered them to be equivalently 'best'. Therefore, we randomly selected one circuit from the group for further comparison and analysis. A sample circuit from this group represents the best tradeoff between performance and sparsity.

## 2.3 CIRCUIT FUNCTIONALITY DISCOVERY

We explore the intended functionalities of circuit components using a combination of attention heat maps and the logit lens. Unlike previous works, we conceptualize the mechanism within the model as either reducing the logits of incorrect choices or enhancing the logits of correct ones. In doing so, we broaden the focus beyond just the target outputs, providing a more comprehensive view of the model's inner mechanism. For IOI-related tasks, we categorize the prompts into ten groups: "B, IO, S1, PLACE, M, S2, V, OBJ, E, and T," following the structure of the IOI prompts. "B" represents the beginning of the sentence (e.g., "When" in Fig. 3), "M" refers to the middle of the sentence (e.g., a comma), and "T" denotes the end of the sentence (e.g., "to"). "V" represents the verb in the second half of the sentence, such as "gave," while "OBJ" refers to the object before "E." "S1" and "S2" refer to the distractor such as "John", while "IO" represents the indirect object the model needs to predict, such as "Mary,". We assign the rest of the templates into "T". For GT tasks, we divide the prompts into six parts: "B, N, V, S, E, and T." "B" represents the beginning of the sentence, "N" refers to the noun that the task focuses on (e.g., "war"), "V" indicates the verb such as "lasts" (suggesting the predicted numbers should be larger), "S" represents the start of the year, and "E" refers to the end of the sentence. We assign the rest of the templates into "T". It is important to note that the intended functionalities do not directly reflect the ultimate contributions of the circuit components to the final output. However, understanding these intended functionalities still provides valuable insights and allows us to estimate their direct effects. As shown in Fig. 3, we found that the previously identified Name Mover Head formed its own group of intended functionality, exhibiting shared similarities in attention patterns. In fact, most of the previously identified nodes with the same functionalities cluster well together when analyzed using the logit lens.

**Attention Grouping** is a technique used to analyze and categorize transformer heads based on their attention patterns. In transformers, each head focuses on different components of the input, reflecting what it "intends" to process. By examining the attention values, we can infer which parts of the sentence each head is attending to and determine its specific intended functionality. Since not all attended tokens carry equal importance, the method focus on the most relevant components. Heads that focus on similar elements, such as an indirect object (IO), are grouped together, suggesting they contribute similarly to the model's decision-making process.

Attention grouping helps uncover the various strategies a model uses to achieve the same outcome. For example, when the indirect object (IO) is emphasized, the model can do this by either increasing attention to the IO or by down-weighting attention to other components. Without grouping heads by their attention patterns, these diverse approaches could be overlooked. By clustering heads that focus on similar components, attention grouping allows for a deeper understanding of how the model processes information and how different heads contribute to the final prediction. This is especially useful in tasks like IOI, where multiple heads may contribute to the same result through distinct ways, offering valuable insights into the model's internal workings.

The attention grouping process involves computing the attention of each head for all tokens in a sentence and mapping them to sentence components (e.g., B, IO). The average attention for each component is calculated, and the top $k$ components with the highest attention values are selected. If a component's attention exceeds the mean across these top components, the head is considered to be attending to that component. Heads are grouped if they attend to the same components. An example can be found in Fig. 3.

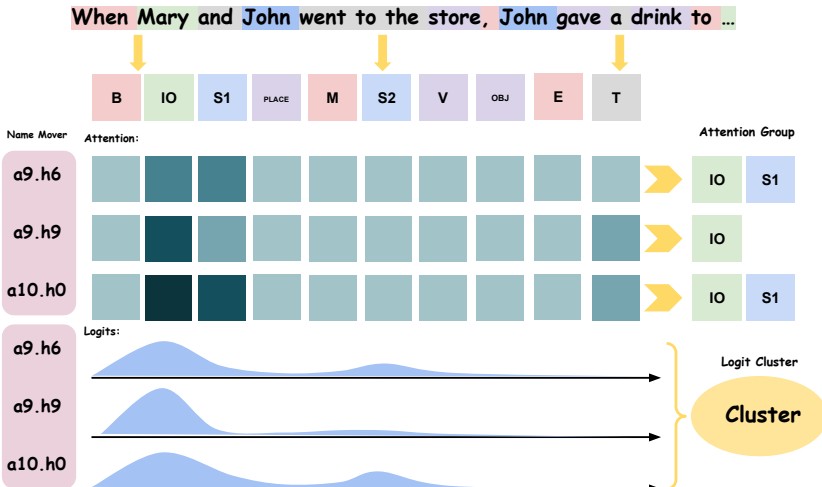

Figure 3: An example of attention grouping and logit clustering from IOI circuit. We assign the IOI prompts into several categories following their general pattern. We found that the previously identified Name Remover Head form its own cluster where the logits on IO get highly up-weighted over the other components. These heads' attention focus on IO and S1/S2 as well, indicating their intentional functionality of attending to and up-weighting IO for correct prediction.

For the IOI task, with 9 sentence components, $k = 5$ is chosen based on observations that heads primarily focus on 4 or less components. For tasks with a different number of sentence components, the value of $k$ can be adjusted accordingly.

**Logit Clustering** is an essential technique used to group transformer heads based on their logit patterns, providing insights into how different heads intended to contribute to the model's decision-making process. One key method used in logit clustering is the *logit lens*, which allows researchers to probe intermediate representations in neural networks (NNs) and transformers. The logit lens works by examining the logits produced at each layer, representing the model's confidence about which token it might output at that specific stage. This technique reveals how early layers begin forming predictions and how these predictions evolve and refine as more layers process the input.

The *logit patterns* generated by different heads give a sense of how each head contributes to the model's predictions by up-weighting or down-weighting specific components. For example, in tasks such as the IOI task, where the indirect object (IO) is the correct target and the first subject (S1) is a distractor, a head that increases the logit on IO while decreasing the logit on S1 plays a crucial role in making the correct prediction. Similarly, other heads may contribute by focusing on irrelevant tokens such as punctuation or template words, which might help the model understand sentence structure or task objectives. Comparing the logit patterns across different heads and layers, thus, helps identify where significant changes in token predictions occur and how intermediate layers potentially contribute to the final outcome.

By clustering heads based on their logit patterns, we can group heads that are performing similar functions. This *logit clustering* allows for the identification of heads that collaborate in the model's decision-making process, providing a clearer view of the functional roles played by different heads. The use of logit clustering is particularly useful when logit patterns vary across layers or even repeat in heads across multiple layers, as we have observed. Applying clustering algorithms to these patterns automates the grouping process, facilitating the analysis of head functions. For example, clustering can reveal whether heads within certain layers, such as deeper layers, are grouped together or whether logit patterns are distributed across all layers.

In this work, we used K-means algorithm for logits clustering. While other clustering methods or human calibrations may provide more precise results, K-means has proven to deliver satisfying results in grouping logit patterns, reducing the amount of manual effort required in functional analysis. This method allows for a scalable approach to understanding how different transformer heads influence the model's output and decisions.

**Attention Grouping + Logit Clustering:** Both attention grouping and logit clustering are needed because each method has limitations when used in isolation. Attention grouping only indicates which components the heads focus on but doesn't reveal whether the heads are attempting to up-weight or down-weight those components. On the other hand, logit clustering identifies which components the heads are likely up-weighting or down-weighting but cannot clarify whether this effect is intentional or a byproduct from another function. For example, an up-weighted logit on IO might either reflect direct attention and up-weighting on this component, or it could be the result of down-weighting other components.

By integrating these two methods, we can better understand the intentional functionality of the heads. Attention grouping helps reveal the intention behind the model's focus, while logit clustering estimates the functionality — the effect the heads are having on specific components. When observing attention groups within the same logit cluster, it's easier to discern how nodes with similar functions differ in their intentions. Similarly, identical attention groups may belong to different logit clusters, indicating varied roles in the model's decision-making process. This combined framework simplifies the investigation of intentional functionalities, helping to clarify how different components intended to contribute to the overall behavior of the model.

## 3 EXPERIMENTAL RESULTS

In this section, we provide a detailed analysis of how the structure and intended functionality of IOI circuits differ from those retrieved from ablated prompts, where we focus on the animal objects since the original GPT2-small is incapable of performing IOI tasks when changing IO and S to cities and colors. We then explore how various finetuning methods enhance performance on both the IOI and GT tasks. Lastly, we investigate how finetuning on the original IOI task improves the model's capability of performing the IOI task when applied to ablated prompts. Evaluations on model performance can be found in Appendix. Sec. B. For the following results, we focus on attention head only by collapsing the QKV nodes into the corresponding attention heads.

Table 1: The model's circuit differences on the GT and IOI tasks are compared in terms of the number of nodes. For clarity, we have consolidated the QKV (Query, Key, Value) nodes of the IOI task into a single attention head. The results are averaged over the selected group of the "best" circuits for each model under each task. A 95% confidence interval is calculated to demonstrate statistically significant variations in circuit sizes. Blocks highlighted in green indicate a statistically significant larger number of nodes. Our analysis shows that finetuning methods generally reduce the circuit sizes for the GT task while introducing more components to the IOI task.

| Model A | Model B | GT task | | | IOI task | | |
| --- | --- | --- | --- | --- | --- | --- | --- |
| | | A only | B only | Shared | A only | B only | Shared |
| Original | IA3 | $17.2 \pm 1.22$ | $11.2 \pm 2.10$ | $38.6 \pm 2.56$ | $16.2 \pm 1.83$ | $20.8 \pm 1.99$ | $74.2 \pm 1.34$ |
| Original | AdaLoRA | $20.0 \pm 1.69$ | $13.8 \pm 1.72$ | $35.8 \pm 2.01$ | $13.0 \pm 2.13$ | $18.2 \pm 1.09$ | $77.4 \pm 1.93$ |
| Original | Bitfit | $25.0 \pm 2.43$ | $11.0 \pm 1.82$ | $30.8 \pm 2.98$ | $21.2 \pm 1.58$ | $20.2 \pm 0.94$ | $69.2 \pm 1.65$ |
| Original | Full | $19.2 \pm 1.72$ | $8.2 \pm 1.78$ | $36.6 \pm 2.86$ | $18.0 \pm 1.84$ | $22.0 \pm 1.07$ | $72.4 \pm 1.37$ |
| Original | LoRA | $20.2 \pm 1.74$ | $14.0 \pm 2.69$ | $35.6 \pm 1.37$ | $14.8 \pm 2.33$ | $22.0 \pm 1.96$ | $75.6 \pm 1.48$ |
| IA3 | AdaLoRA | $10.4 \pm 1.63$ | $10.2 \pm 1.28$ | $39.4 \pm 2.36$ | $17.8 \pm 1.22$ | $18.4 \pm 1.23$ | $77.2 \pm 1.01$ |
| IA3 | Bitfit | $17.8 \pm 2.87$ | $9.8 \pm 1.74$ | $32.0 \pm 3.05$ | $22.8 \pm 1.91$ | $15.4 \pm 1.13$ | $72.2 \pm 1.40$ |
| IA3 | Full | $13.0 \pm 2.46$ | $8.0 \pm 2.57$ | $36.8 \pm 1.95$ | $19.2 \pm 0.94$ | $18.6 \pm 1.29$ | $75.8 \pm 0.58$ |
| IA3 | LoRA | $10.8 \pm 2.37$ | $10.6 \pm 2.07$ | $39.0 \pm 1.57$ | $17.0 \pm 1.07$ | $19.6 \pm 1.43$ | $78.0 \pm 1.04$ |
| LoRA | AdaLoRA | $9.6 \pm 1.81$ | $9.6 \pm 1.58$ | $40.0 \pm 1.90$ | $17.6 \pm 1.35$ | $15.6 \pm 1.99$ | $80.0 \pm 1.33$ |
| LoRA | Bitfit | $18.0 \pm 2.16$ | $10.2 \pm 2.99$ | $31.6 \pm 1.85$ | $22.4 \pm 2.68$ | $12.4 \pm 1.09$ | $75.2 \pm 1.01$ |
| LoRA | Full | $14.0 \pm 2.35$ | $9.2 \pm 2.85$ | $35.6 \pm 1.72$ | $19.2 \pm 1.60$ | $16.0 \pm 1.07$ | $78.4 \pm 0.90$ |
| AdaLoRA | Bitfit | $17.4 \pm 1.81$ | $9.6 \pm 1.97$ | $32.2 \pm 2.87$ | $22.0 \pm 0.78$ | $14.0 \pm 1.21$ | $73.6 \pm 0.66$ |
| AdaLoRA | Full | $14.4 \pm 1.56$ | $9.6 \pm 2.03$ | $35.2 \pm 2.55$ | $20.2 \pm 1.22$ | $19.0 \pm 1.21$ | $75.4 \pm 0.53$ |
| Bitfit | Full | $11.2 \pm 2.39$ | $14.2 \pm 1.95$ | $30.6 \pm 2.89$ | $14.6 \pm 1.72$ | $21.4 \pm 1.85$ | $73.0 \pm 1.07$ |

### 3.1 IOI CIRCUITS AND ABLATED PROMPTS

As shown in Fig. 1, substituting human names with other types of objects, such as animals, leads to significantly smaller circuits. Additionally, as illustrated in Fig. 4, we observed three distinct

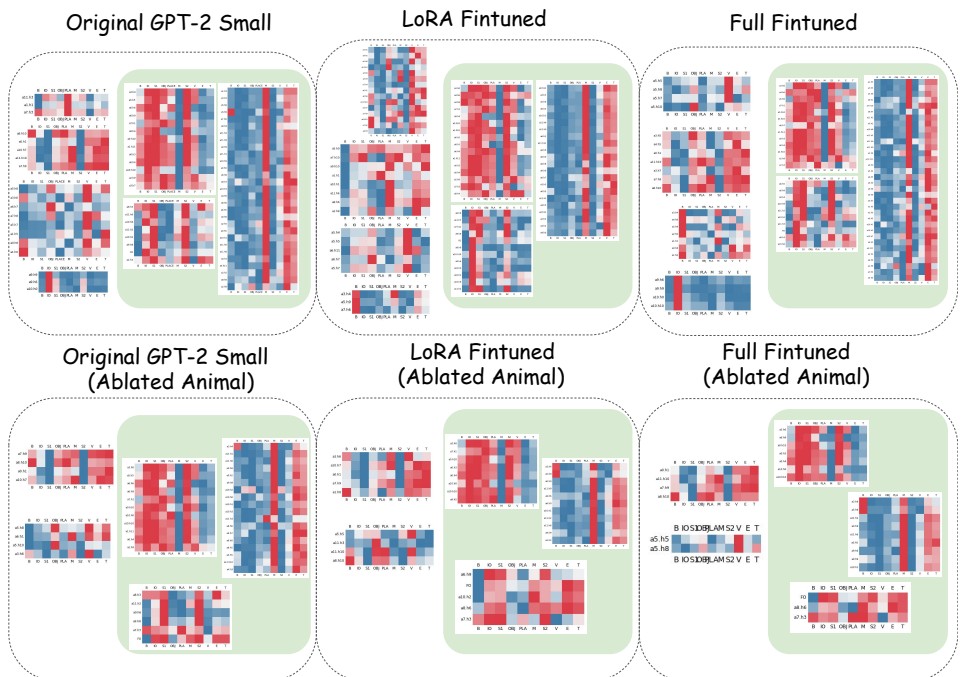

Figure 4: Logit clusters across fine-tuning methods and ablated prompts. The clusters are mostly uniform across fine-tuning methods for either prompt setting, indicating that fine-tuning does not significantly modify circuit's main functionalities; however, variations can be observed, both in terms of the general sizes of each cluster and individual head's intentional functionalities. Comparing IOI and ablated prompts, the ablation circuits consist of fewer functional clusters than IOI, and the number of heads for the shared clusters generally reduces. This is likely due to the complexity of processing human names in a semantically meaningful sentence which requires more functional nodes.

clustering groups that consistently appear across all circuits in both IOI and IOI-ablated tasks. One of the clusters places a strong emphasis on the M, E, and T parts of the sentence, in terms of logit, while also attending to the entire sentence. We assume that these nodes intend to complete the sentence with the simplest solutions, such as adding a comma or repeating the final word. Another cluster, in contrast, emphasizes all parts of the sentence except for M, E, and T, while still attending to the entire sentence. This cluster appears to function in opposition to the sentence completion cluster. This cluster also aligns with where all previously identified induction heads (Wang et al., 2022) are clustered. The final preserved cluster focuses heavily on the IO and S tokens, both in terms of logits and attention. This cluster generally down-weights OBJECT and PLACE, components that are irrelevant to correct prediction, which makes it distinguished from the induction heads cluster who usually up-weights them.

In terms of the general cluster analysis, the cluster of induction heads are mostly preserved, with only three additional nodes in IOI. The sentence completion cluster has significantly more heads in IOI, contributing to the difference in circuit sparsity. The cluster that focuses on IO and S is mostly preserved. S-inhibition and the negative name mover (downweighting IO and S, upweighting others) are also preserved.

Interestingly, there is no cluster found in the animals-ablated circuit that purely concentrates on IO in terms of logits and attention. It is worth noting that most of the attention patterns from ablated prompts do not involve any exclusive focus on IO or S tokens in terms of logits and attentions. Instead, they tend to focus on IO and S along with the beginning words. Furthermore, the name mover heads, which are well-clustered in the IOI circuits shown in Fig. 3, now appear to be more dispersed. For instance, node 10.0 still upweights IO over S1, but the effect is less pronounced. It also shows a strong upweighting of PLACE. Node 9.6 shifts to upweight S over IO, while node 9.9 joins the

sentence completion cluster, upweighting M, E, and T but downweighting the other elements. These findings overall suggest that the model struggles to isolate IO and S from the prompt. Most of the time, it relies on the beginning of the sentence to identify the IO and S pair. Although the behavior of indirect object identification persists, the model's limited ability to accurately identify IO and S hinders its performance compared to the original IOI task. These findings are consistent across finetuning methods and ablation settings.

## 3.2 PEFT METHODS COMPARISON

By fine-tuning GPT-2 small on IOI and GT tasks, we observed that the circuits retrieved from GT tasks are generally statistically significantly smaller than those from the original GT tasks, as shown in Table 1. An analysis of logits and attention patterns on GT revealed that the fine-tuned model tends to focus more on the starting year, while the MLP in deeper layers increasingly up-weights target number continuations. These findings suggest that the reduction in circuit size for GT tasks is likely due to the simplicity of the task —simply selecting numbers greater than the starting year, as the task only has one direction (Greater Than)— and a more focused approach in solving it. More results on GT tasks can be found in Appendix. Sec. D. In contrast, for the more complex IOI task, fine-tuning methods generally result in a statistically significantly larger circuit to address the task's increased complexity.

For the IOI task, the overall structure of functionality for circuits is largely preserved, aligning with findings from Prakash et al.. Specifically, many of the heads identified by Wang et al. as serving certain functions continue to perform similar roles in fine-tuned cases. For example, Duplicate Token Heads, Previous Token Heads, some S-inhibition Heads, and certain Backup Name Mover Heads remain consistent across all fine-tuning methods, maintaining their functionalities despite the fine-tuning adjustments. This stability in key components ensures that model behavior is preserved, allowing fine-tuned heads to focus more effectively on specific task elements, thereby enhancing overall task performance. Furthermore, this consistency not only highlights the critical contribution of these heads to solving the IOI task but also validates our framework, which integrates attention grouping and logit clustering to explore the intended functionalities of LLM attention heads.

While many individual heads' intentional functionalities are generally preserved, fine-tuning often causes shifts in their behavior. Different fine-tuning methods may or may not achieve the same effect on this change in behavior. For instance, in the two Induction Heads identified by Wang et al., a5.h5 and a6.h9, we observe that while fine-tuning preserves the intentional functionality of a6.h9, a5.h5 undergoes significant modifications with different fine-tuning methods. In the original circuit, a5.h5 attends to the beginning of the sentence, up-weighting B, IO, S, OBJ, and down-weighting M, E, and T. However, after fine-tuning, Full Finetune, BitFit, and LoRA shift it to primarily up-weight V, whereas AdaLoRA and IA3 reverse the effect, down-weighting B, IO, S, OBJ, and PLACE, while up-weighting M, V, E, and T. Additional example of how fine-tuning methods align or differ in modifying the Name Mover Heads on IOI circuits is shown in Fig. 5.

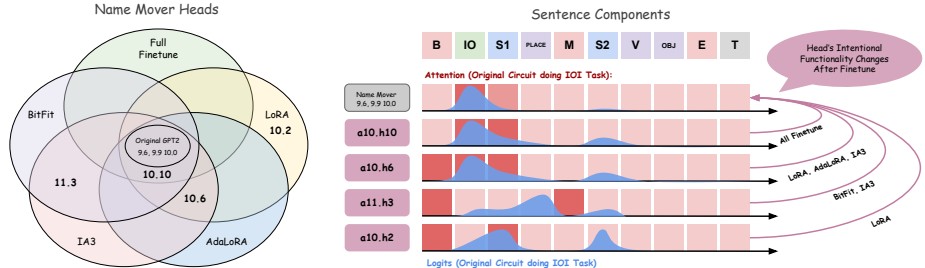

Figure 5: A comparison of how different fine-tuning methods modify the cluster of Name Mover Heads claimed by prior works. The Name Mover Heads significantly attend to and up-weight IO over other components. All fine-tuning methods increase the number of these heads to enhance prediction accuracy, shifting the functionalities of some particular heads in the original GPT-2 circuit for better performance. While some fine-tuning methods share similarities in the additional Name Mover Heads, variations are common.

Therefore, the differences between circuits may help explain how fine-tuning enhances the performance of LLMs on certain tasks. From our observations, fine-tuning increases the number of heads that up-weight logits on IO compared to the original GPT-2 circuit. Specifically, the improvement is primarily due to more "focused" up-weighting on IO relative to other components. As shown in Fig. 5, all circuits from fine-tuned models enforce certain heads from the original circuit to function similarly to Name Mover Heads—those that primarily attend to and up-weight IO—leading to improved task performance. Notably, all fine-tuning methods modify the functionality of a10.h10, likely due to its high similarity to Name Mover Heads. However, PEFT requires a larger number of functionality-shifted heads to achieve comparable performance, which is unsurprising since full fine-tuning updates all weights and biases, thoroughly converting the functionalities of individual heads. In contrast, PEFT only updates a small subset of parameters, necessitating more heads with similar intentional functionalities to match the performance of fully fine-tuned circuits.

In conclusion, both PEFT methods and full fine-tuning improve performance not only by increasing the presence of highly specialized heads, such as Name Mover and Induction Heads, but also by simplifying and refining circuits through the pruning of irrelevant or less important components. This dual effect—enhancing the specificity and accuracy of critical heads while reducing unnecessary complexity—highlights the vital role of fine-tuning in optimizing circuit performance for specific tasks. These adjustments lead to more efficient and interpretable circuits that maintain high levels of task-specific accuracy.

## 4 CONCLUSIONS AND LIMITATIONS

Overall, our framework significantly reduces the reliance on manual efforts, enhancing the efficiency of discovering the intended functionality of circuit components. Through our analysis of how the circuits for the IOI and GT tasks vary across different fine-tuning methods and ablated prompts, we found that while the overall structure of intended functionality is preserved, the specific components responsible for these functions may change. This finding suggests that pre-identified circuits and functionalities are subject to variations depending on the ablated prompts and fine-tuning methods used. We hope that our findings and framework will inspire further exploration of circuit utilization and its interpretability in LLMs.

As discussed in the paper, rather than estimating the direct effect of individual nodes on target logit values, we focus on their intended functionalities. While exploring intended functionalities can provide a reasonable approximation of direct effects—since pre-identified heads with similar functions are generally well-clustered—it is important to recognize that they are not equivalent. Drawing an analogy to "correlation is not causation," we emphasize that intended functionality does not necessarily reflect the "true" functionality of circuit components. Furthermore, intended functionality may depend on the performance of previous states, meaning it can become inconsistent if those earlier states are perturbed. However, this challenge could also arise in causal exploration methods like Patch patching.

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

# APPENDIX TABLE OF CONTENTS

## A    MODEL PERFORMANCE

## B    BACKGROUND AND RELATED WORKS

**Mechanistic Interpretability** (MI) (Elhage et al., 2021; Olsson et al., 2022) aims to discover interpretable components of an otherwise blackbox neural network. Such analyses are typically performed via a series of input perturbations (also known as *patching*) to ablate the effect of individual model components to its predictive behavior (Chan et al., 2022; Meng et al., 2022a;b; Goldowsky-Dill et al., 2023). MI has been successfully applied to natural language tasks that feature a controlled output space on small-scale pre-trained LMs (Wang et al., 2022; Hanna et al., 2024); it has also been applied to study in-context learning and algorithmic behaviors on stylized transformers (Akyürek et al., 2022; Fu et al., 2023; Nanda et al., 2023). While early works put equal focus on circuit discovery and interpretability, recent ones have emphasized scalable circuit discovery (Conmy et al., 2023; Syed et al., 2023; Bhaskar et al., 2024) over extracting human-understandable algorithms, which requires extensive manual effort to examine the computational behavior of model components. Our work integrates recent best practices in circuit discovery to *automate* interpretability.

**Parameter-Efficient Fine-tuning** (Mangrulkar et al., 2022) aims to improve model performance on downstream tasks by training only a small portion of parameter relative to the full model. Referring to Ding et al. (2023) for a more detailed survey, most popular approaches exploits the low-rank structure of projection matrices (Hu et al., 2021; Zhang et al., 2023) or introduce a fixed set of scaling and/or bias parameters (Zaken et al., 2021; Liu et al., 2022). Other representative approaches include prompt tuning (Lester et al., 2021; Li & Liang, 2021; Diao et al., 2022) and dynamically identifying tuning parameters via influence functions (Sung et al., 2021). In this work, we primarily investigate LoRA (Hu et al., 2021), AdaLora (Zhang et al., 2023), BitFit (Zaken et al., 2021), and IA3 (Liu et al., 2022). These methods are broadly applied in many production settings, since they are more scalable with commercial hardware and can be served with thousands of replicas simultaneously (Sheng et al., 2023).

Bhaskar et al. has studied the change effects of circuits induced by fine-tuning on an entity tracking task, and found that fine-tuning *enhances* existing mechanisms for billion-scale LMs. Our work extends this study with a more diverse set of tasks and PEFT methods; and more importantly, we have identified that fine-tuning *modifies* existing mechanisms for small LMs.

This section presents the evaluation results of all models across various tasks and ablated prompts. Although the model performs well on the IOI ablated tasks after fine-tuning, GPT-2 small struggles with indirect object identification when the main objects are replaced with colors and cities. Specifically, the model's ability to handle these ablations remains limited, indicating that fine-tuning has not fully generalized the model to different types of input modifications. Moreover, even after fine-tuning, the performance on the IOI task ablated with cities is still suboptimal, suggesting that the model's understanding of abstract entities such as locations remains insufficient. These findings highlight the need for more targeted interventions or further fine-tuning strategies to improve model robustness across diverse ablations.

## C    IOI CIRCUITS ANALYSIS

This section documents the detailed qualitative results for IOI analysis. Sec. C.1 lists the visualization of comparative circuit analysis. The overlapped nodes are in white colors while the unique nodes to each cicuits are shown in either pink or blue. Sec. C.2 lists the results of logit clustering and attention group visualizations.

### C.1    CIRCUIT COMPARISON

Comparative circuit analysis on the original IOI tasks can be found in Fig. 6, Fig. 7, Fig. 8,Fig. 9, Fig. 10, and Fig. 11. The analysis with ablated prompts with animals can be found in Fig. 12, Fig. 13, Fig. 14, and Fig. 17. The analysis with ableted prompts with cities can be found in Fig. 18 and Fig. 19. Analysis on prompts ablated with colors can be found in Fig. 21.

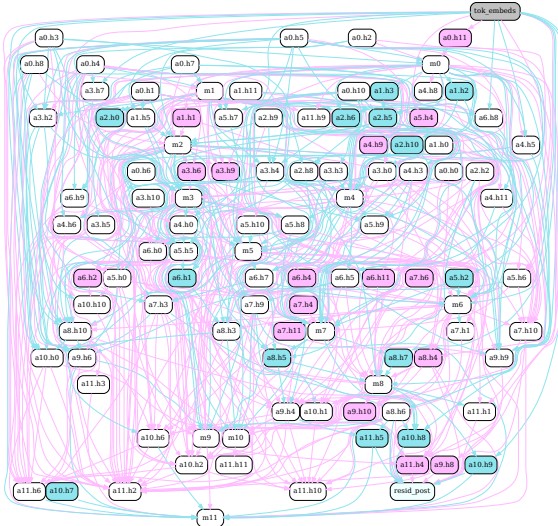

Figure 6: Circuit difference: LoRA vs Full. Red/pink indicates LoRA-specific circuits, blue indicates Full-specific circuits, and white represents shared circuits.

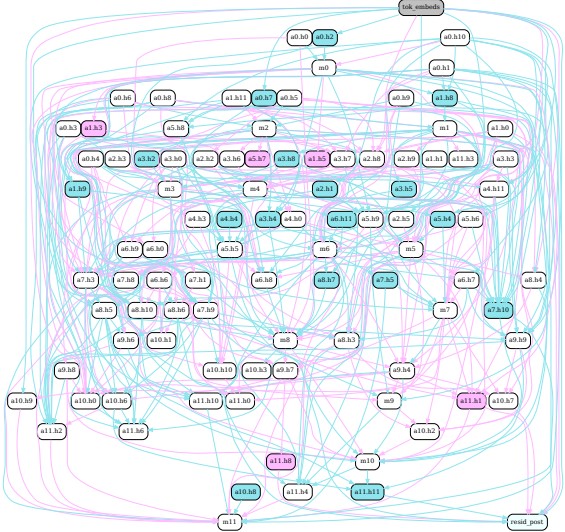

Figure 7: Circuit difference: Original vs AdaLoRA. Red/pink indicates Original-specific circuits, blue indicates AdaLoRA-specific circuits, and white represents shared circuits.

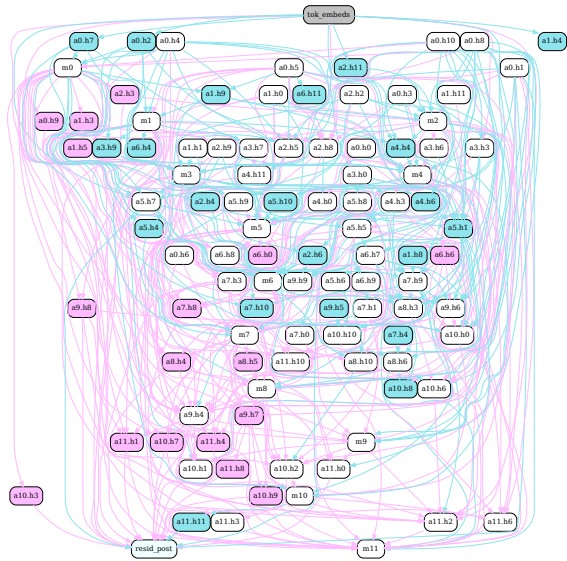

Figure 8: Circuit difference: Original vs BitFit. Red/pink indicates Original-specific circuits, blue indicates BitFit-specific circuits, and white represents shared circuits.

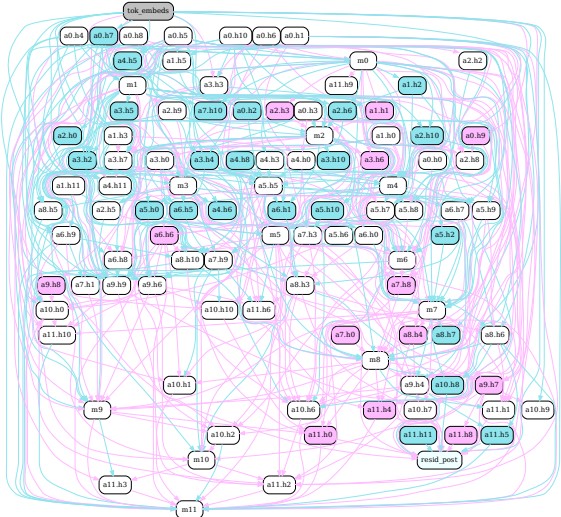

Figure 9: Circuit difference: Original vs Full. Red/pink indicates Original-specific circuits, blue indicates Full-specific circuits, and white represents shared circuits.

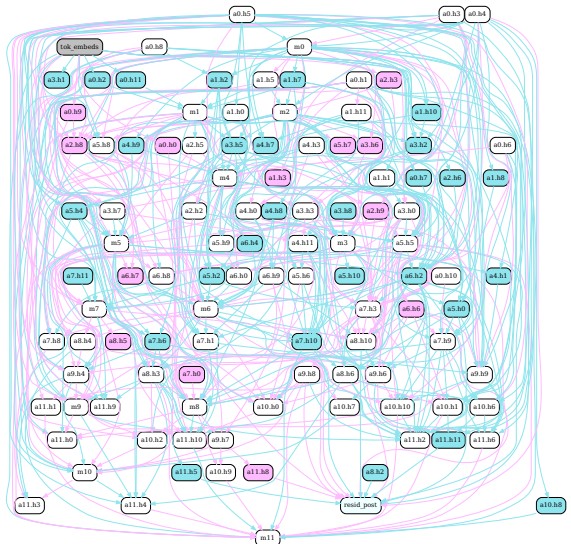

Figure 10: Circuit difference: Original vs IA3. Red/pink indicates Original-specific circuits, blue indicates IA3-specific circuits, and white represents shared circuits.

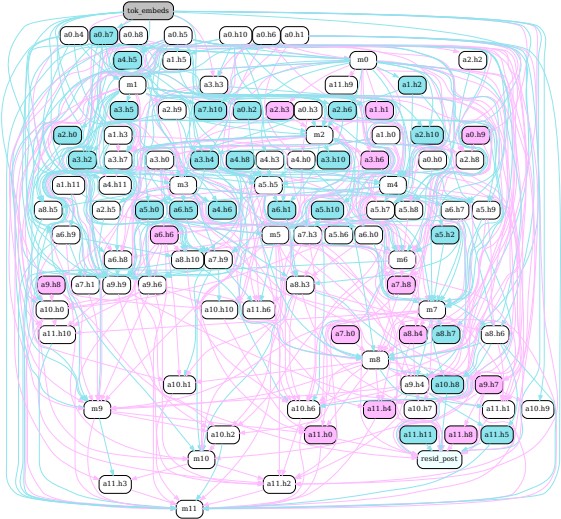

Figure 11: Circuit difference: Original vs LoRA. Red/pink indicates Original-specific circuits, blue indicates LoRA-specific circuits, and white represents shared circuits.

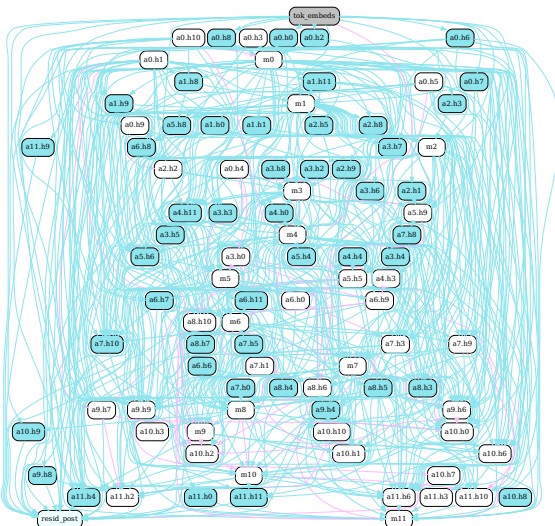

Figure 12: AdaLoRA circuit difference between IOI and IOI-animals, where blue corresponds to IOI only and red corresponds to IOI-animals only circuit components. White refers to shared circuit components.

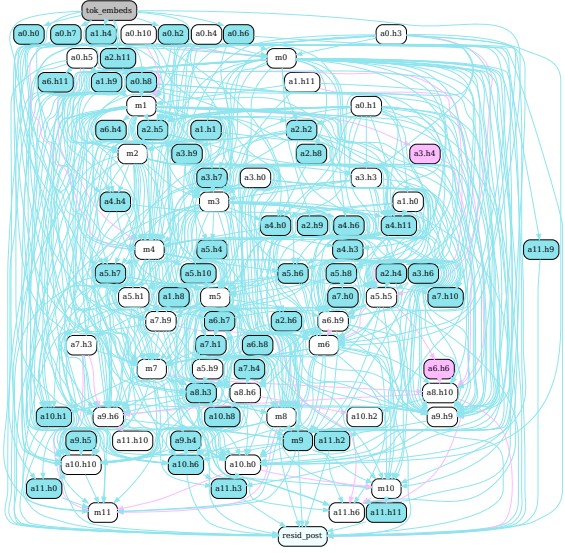

Figure 13: BitFit circuit difference between IOI and IOI-animals, where blue corresponds to IOI only and red corresponds to IOI-animals only circuit components. White refers to shared circuit components.

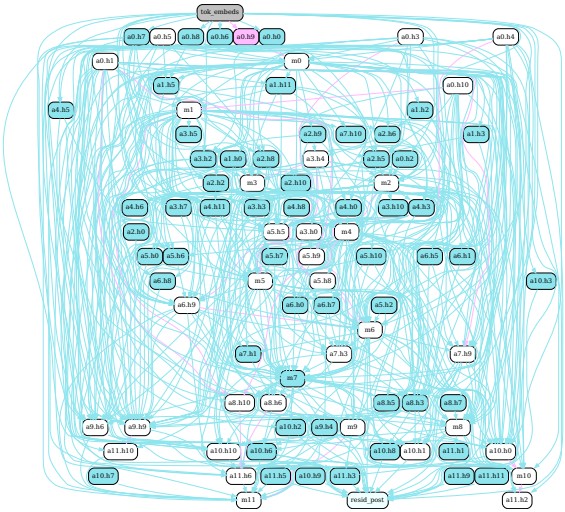

Figure 14: Full circuit difference between IOI and IOI-animals, where blue corresponds to IOI only and red corresponds to IOI-animals only circuit components. White refers to shared circuit components.

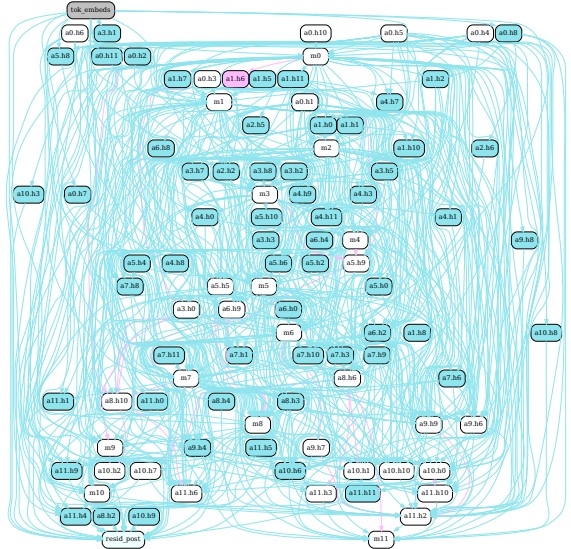

Figure 15: IA3 circuit difference between IOI and IOI-animals, where blue corresponds to IOI only and red corresponds to IOI-animals only circuit components. White refers to shared circuit components.

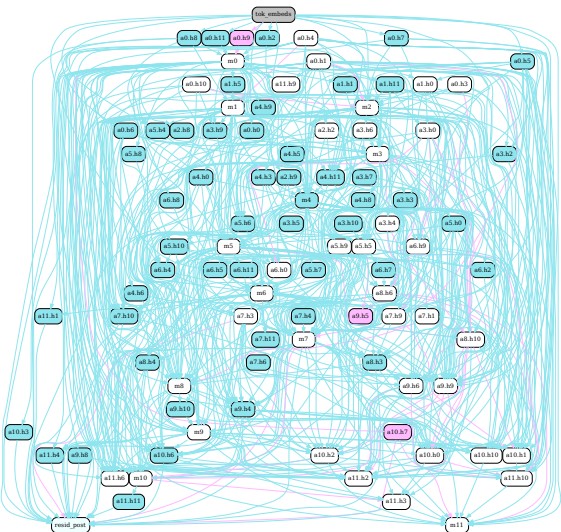

Figure 16: LoRA circuit difference between IOI and IOI-animals, where blue corresponds to IOI only and red corresponds to IOI-animals only circuit components. White refers to shared circuit components.

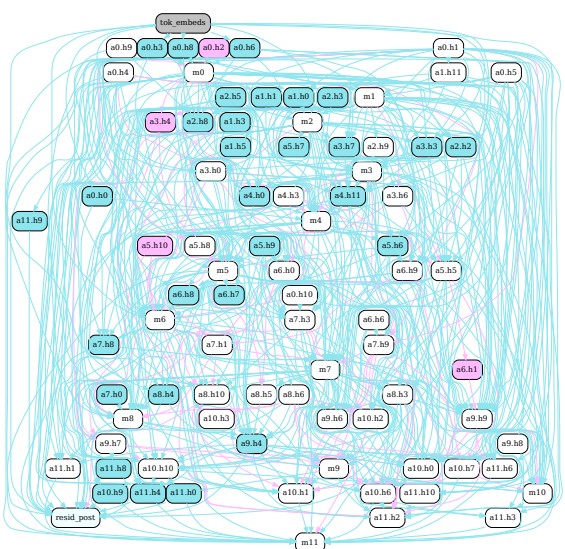

Figure 17: Original circuit difference between IOI and IOI-animals, where blue corresponds to IOI only and red corresponds to IOI-animals only circuit components. White refers to shared circuit components.

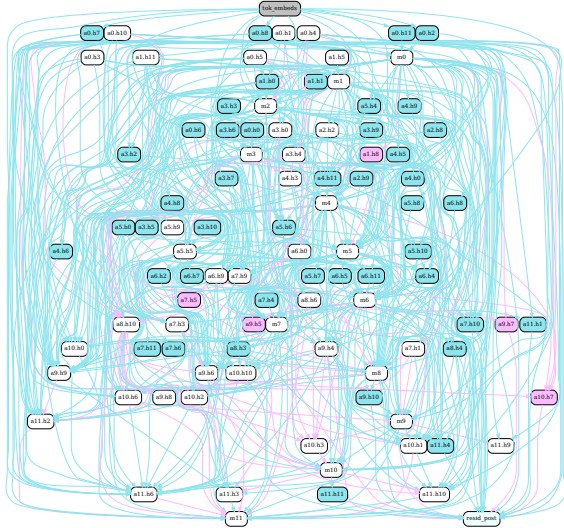

Figure 18: LoRA circuit difference between IOI and IOI-cities, where blue corresponds to IOI only and red corresponds to IOI-cities only circuit components. White refers to shared circuit components.

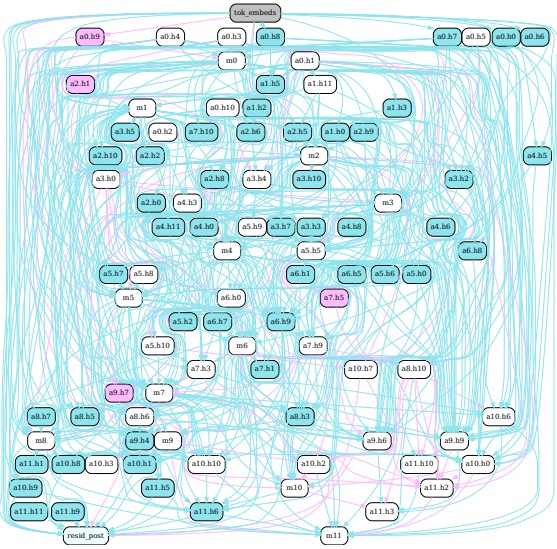

Figure 19: Full finetune circuit difference between IOI and IOI-cities, where blue corresponds to IOI only and red corresponds to IOI-cities only circuit components. White refers to shared circuit components.

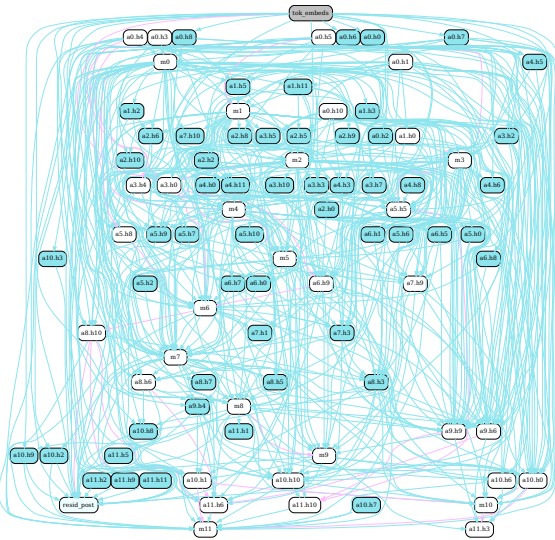

Figure 20: LoRA circuit difference between IOI and IOI-colors, where blue corresponds to IOI only and red corresponds to IOI-colors only circuit components. White refers to shared circuit components.

Figure 21: Full finetune circuit difference between IOI and IOI-colors, where blue corresponds to IOI only and red corresponds to IOI-colors only circuit components. White refers to shared circuit components.

Table 2: IOI circuit performances, rounded to the nearest hundredth. The original GPT-2 model, due to poor accuracy on colors and cities ablated settings, is not satisfiable for meaningful circuit investigations on these two tasks. Among fine-tuning methods, cities-ablated prompts pose the most difficulty for the circuit to preserve full model performance.

| Metric | Ablation | AdaLoRA | BitFit | Full | IA3 | LoRA | Original |
|---|---|---|---|---|---|---|---|
| Acc | Names | 0.95 ± 0.01 | 0.99 ± 0.00 | 0.98 ± 0.00 | 0.97 ± 0.00 | 0.94 ± 0.00 | 0.73 ± 0.01 |
| | Animals | 0.96 ± 0.02 | 0.99 ± 0.00 | 0.99 ± 0.00 | 0.96 ± 0.01 | 0.97 ± 0.00 | 0.52 ± 0.01 |
| | Colors | 0.95 ± 0.02 | 0.99 ± 0.00 | 0.99 ± 0.00 | 0.96 ± 0.01 | 0.96 ± 0.00 | – |
| | Cities | 0.74 ± 0.02 | 0.97 ± 0.01 | 0.93 ± 0.01 | 0.76 ± 0.02 | 0.79 ± 0.00 | – |
| LD | Names | 5.95 ± 0.16 | 11.62 ± 0.15 | 11.63 ± 0.12 | 6.15 ± 0.10 | 5.83 ± 0.12 | 3.24 ± 0.18 |
| | Animals | 4.61 ± 0.41 | 12.32 ± 0.43 | 11.03 ± 0.34 | 4.35 ± 0.30 | 5.17 ± 0.27 | 0.86 ± 0.07 |
| | Colors | 4.23 ± 0.42 | 11.50 ± 0.39 | 10.41 ± 0.36 | 4.39 ± 0.31 | 4.91 ± 0.20 | – |
| | Cities | 2.57 ± 0.20 | 9.60 ± 0.38 | 8.82 ± 0.34 | 2.19 ± 0.15 | 3.37 ± 0.08 | – |
| KL | Names | 0.21 ± 0.01 | 0.05 ± 0.00 | 0.06 ± 0.01 | 0.16 ± 0.01 | 0.20 ± 0.01 | 0.30 ± 0.00 |
| | Animals | 0.09 ± 0.02 | 0.06 ± 0.01 | 0.04 ± 0.01 | 0.08 ± 0.01 | 0.06 ± 0.01 | 0.28 ± 0.01 |
| | Colors | 0.10 ± 0.03 | 0.05 ± 0.01 | 0.04 ± 0.00 | 0.09 ± 0.02 | 0.07 ± 0.00 | – |
| | Cities | 0.38 ± 0.04 | 0.11 ± 0.02 | 0.21 ± 0.03 | 0.39 ± 0.03 | 0.37 ± 0.01 | – |
| EM | Names | 0.95 ± 0.00 | 0.99 ± 0.00 | 0.98 ± 0.00 | 0.96 ± 0.00 | 0.94 ± 0.00 | 0.76 ± 0.01 |
| | Animals | 0.96 ± 0.02 | 0.99 ± 0.00 | 0.99 ± 0.00 | 0.96 ± 0.01 | 0.97 ± 0.00 | 0.66 ± 0.01 |
| | Colors | 0.95 ± 0.02 | 0.99 ± 0.00 | 0.99 ± 0.00 | 0.96 ± 0.01 | 0.97 ± 0.00 | – |
| | Cities | 0.77 ± 0.02 | 0.97 ± 0.01 | 0.93 ± 0.01 | 0.77 ± 0.02 | 0.82 ± 0.00 | – |

Table 3: GT task performances

| Finetune Method | ES | PD | PD(10) | KT | KL |
|---|---|---|---|---|---|
| Original | 0.99 ± 0.00 | 0.72 ± 0.00 | 0.33 ± 0.01 | 0.78 ± 0.01 | 0.23 ± 0.02 |
| AdaLoRA | 0.99 ± 0.00 | 0.96 ± 0.00 | 0.62 ± 0.01 | 0.84 ± 0.01 | 0.38 ± 0.02 |
| BitFit | 0.99 ± 0.00 | 1.00 ± 0.00 | 0.02 ± 0.00 | 0.88 ± 0.00 | 0.20 ± 0.01 |
| Full | 0.99 ± 0.00 | 0.98 ± 0.00 | 0.53 ± 0.03 | 0.82 ± 0.01 | 0.74 ± 0.05 |
| IA3 | 0.99 ± 0.00 | 0.92 ± 0.00 | 0.55 ± 0.01 | 0.83 ± 0.00 | 0.33 ± 0.01 |
| LoRA | 0.99 ± 0.00 | 0.92 ± 0.00 | 0.58 ± 0.01 | 0.83 ± 0.01 | 0.30 ± 0.02 |

## C.2  LOGIT CLUSTERING AND ATTENTION GROUPING

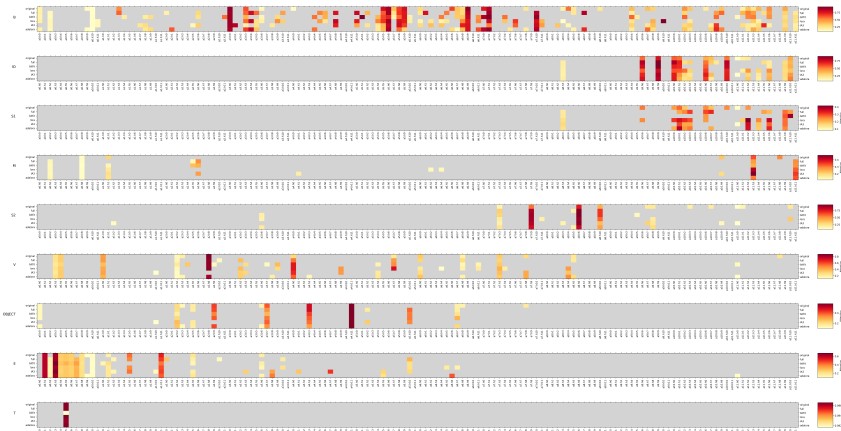

Figure 22: Attention grouping results for all components circuit. Grey cells indicate pruned nodes of minimal attention.

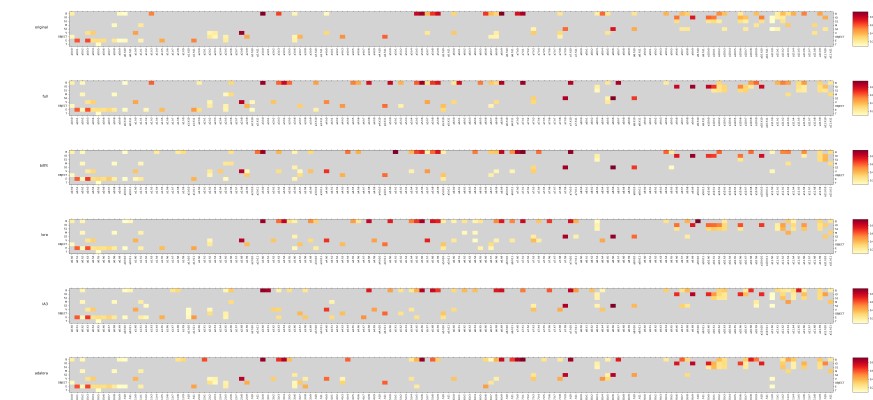

Figure 23: Attention grouping results for all finetuning methods circuit. Grey cells indicate pruned nodes of minimal attention.

# D GT CIRCUIT ANALYSIS

## D.1 CIRCUIT COMPARISON

## D.2 LOGIT CLUSTERING AND ATTENTION GROUPING

# E REPRODUCIBILITY

The implementation of circuit discovery mainly depends on the code from Edge Pruning. More details can be found in their github. The code to perform logit lens and attention grouping will be released upon acceptance.

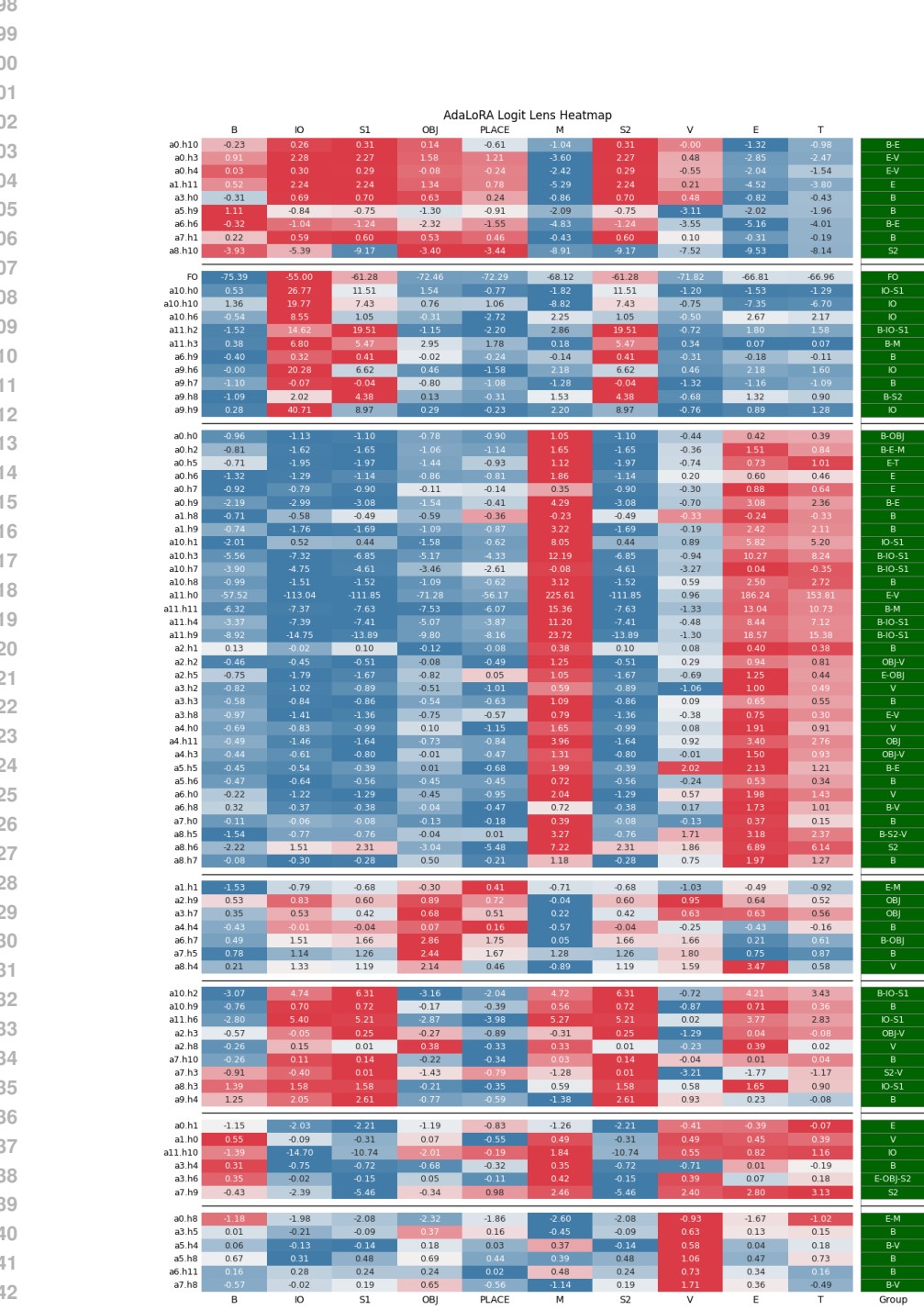

Figure 24: AdaLoRA Logit Clusters

Bitfit Logit Lens Heatmap

| | B | IO | S1 | OBJ | PLACE | M | S2 | V | E | T | |
|---|---|---|---|---|---|---|---|---|---|---|---|
| FO | -65.51 | -40.48 | -51.64 | -59.80 | -60.72 | -56.71 | -51.64 | -58.74 | -55.73 | -55.66 | FO |
| a10.h2 | -2.02 | 1.04 | 1.10 | -1.75 | -1.19 | 3.44 | 1.10 | 0.61 | 2.61 | 2.24 | B |
| a10.h6 | -1.34 | 5.22 | 1.47 | -0.61 | -2.91 | 2.56 | 1.47 | 0.28 | 2.90 | 2.35 | IO-S1 |
| a11.h2 | -0.85 | 2.19 | 3.92 | -0.35 | -0.91 | 2.33 | 3.92 | -0.17 | 1.84 | 1.27 | B |
| a3.h0 | -0.39 | 0.32 | 0.33 | 0.36 | 0.00 | 0.16 | 0.33 | 0.45 | -0.23 | 0.13 | B |
| a5.h1 | -0.50 | 0.03 | -0.08 | -0.25 | -0.45 | 0.11 | -0.08 | -0.46 | 0.15 | -0.01 | B |
| a6.h11 | 0.09 | 0.51 | 0.42 | 0.49 | 0.19 | 0.39 | 0.42 | 0.63 | 0.64 | 0.16 | B-V |
| a7.h0 | 0.03 | 0.18 | 0.18 | 0.04 | -0.16 | 0.04 | 0.18 | -0.03 | 0.18 | 0.03 | B |
| a7.h10 | -0.06 | 0.04 | 0.05 | -0.04 | -0.28 | 0.07 | 0.05 | 0.16 | 0.21 | 0.11 | B |
| a8.h3 | 1.35 | 1.96 | 1.62 | -0.80 | -1.67 | 0.17 | 1.62 | 0.64 | 1.54 | 0.71 | IO-S1 |
| a8.h6 | -2.78 | 0.90 | 1.25 | -4.12 | -6.58 | 10.19 | 1.25 | 0.30 | 8.43 | 7.42 | S2 |
| a1.h1 | -1.33 | -0.79 | -0.69 | -0.29 | 0.44 | -0.32 | -0.69 | -0.86 | -0.29 | -0.57 | E-M |
| a11.h10 | -2.05 | -3.57 | -2.87 | -0.83 | 1.61 | -1.64 | -2.87 | -1.92 | -1.87 | -1.29 | IO-S1 |
| a2.h6 | -0.64 | -0.68 | -0.77 | -0.21 | -0.00 | -0.04 | -0.77 | -0.64 | 0.14 | -0.34 | B-M |
| a2.h8 | -0.17 | 0.18 | 0.04 | 0.54 | -0.06 | 0.45 | 0.04 | -0.19 | 0.62 | 0.10 | V |
| a3.h7 | 0.37 | 0.32 | 0.21 | 0.53 | 0.46 | 0.43 | 0.21 | 0.64 | 0.93 | 0.66 | OBJ-V |
| a5.h10 | -1.04 | -1.12 | -1.27 | 1.86 | -0.82 | -0.02 | -1.27 | 0.79 | 1.02 | 0.40 | B-E-OBJ |
| a6.h8 | 0.55 | 0.29 | 0.19 | 0.42 | -0.14 | 0.25 | 0.19 | 0.16 | 1.57 | 1.01 | B-V |
| a0.h1 | -1.50 | -2.20 | -2.37 | -1.38 | -0.97 | -1.43 | -2.37 | -0.50 | -0.54 | -0.24 | E |
| a0.h8 | -0.53 | -1.06 | -1.13 | -1.36 | -1.03 | -1.05 | -1.13 | -0.41 | -0.77 | -0.35 | E-M |
| a1.h0 | 0.54 | -0.02 | -0.20 | 0.18 | -0.43 | 0.10 | -0.20 | 0.44 | 0.18 | 0.20 | E-V |
| a3.h6 | 0.38 | -0.11 | -0.24 | -0.01 | -0.19 | 0.23 | -0.24 | 0.31 | -0.05 | 0.05 | B-E-S2 |
| a5.h9 | 0.58 | -1.42 | -1.47 | -1.64 | -0.76 | 0.38 | -1.47 | -1.89 | 0.34 | -0.15 | B |
| a7.h9 | -0.74 | -3.22 | -6.19 | -0.57 | 0.61 | 4.38 | -6.19 | 2.61 | 3.59 | 3.88 | S2 |
| a8.h10 | -2.68 | -7.13 | -11.25 | -3.46 | -2.24 | -4.35 | -11.25 | -4.30 | -3.17 | -3.63 | S2 |
| a0.h0 | -0.59 | -0.88 | -0.83 | -0.60 | -0.71 | 0.71 | -0.83 | -0.20 | 0.40 | 0.35 | B-OBJ |
| a0.h10 | -0.43 | -0.47 | -0.44 | -0.14 | -0.38 | 0.24 | -0.44 | 0.15 | 0.10 | 0.08 | B-E |
| a0.h2 | -0.41 | -1.15 | -1.16 | -0.54 | -0.48 | 1.55 | -1.16 | 0.14 | 1.36 | 1.02 | B-E-M |
| a0.h5 | -0.80 | -1.99 | -2.04 | -1.42 | -0.99 | 1.22 | -2.04 | -0.77 | 0.72 | 0.99 | E-T |
| a0.h6 | -1.73 | -1.90 | -1.90 | -1.47 | -1.30 | 0.78 | -1.90 | -0.31 | 0.01 | 0.03 | E |
| a0.h7 | -0.82 | -0.58 | -0.71 | -0.15 | -0.19 | 0.29 | -0.71 | -0.48 | 0.53 | 0.41 | E |
| a1.h8 | -0.96 | -1.45 | -1.32 | -1.15 | -0.81 | 1.28 | -1.32 | -0.43 | 0.97 | 0.72 | B |
| a1.h9 | -0.65 | -1.64 | -1.64 | -1.12 | -0.99 | 2.49 | -1.64 | -0.27 | 2.07 | 1.72 | B |
| a10.h1 | -1.40 | 2.48 | -0.07 | -1.54 | 0.16 | 7.65 | -0.07 | 2.46 | 5.16 | 5.05 | IO |
| a10.h8 | -3.46 | -4.78 | -4.82 | -3.42 | -2.79 | 8.14 | -4.82 | 0.11 | 5.63 | 5.64 | B |
| a11.h0 | -51.93 | -102.71 | -101.47 | -65.78 | -50.96 | 201.10 | -101.47 | 0.34 | 166.50 | 137.75 | B-V |
| a11.h11 | -7.07 | -8.77 | -8.73 | -6.38 | -6.11 | 11.39 | -8.73 | 0.66 | 9.65 | 8.08 | B-M |
| a11.h6 | -2.94 | -1.04 | -0.56 | -2.97 | -2.88 | 6.90 | -0.56 | 0.84 | 5.45 | 4.33 | B-IO |
| a11.h9 | -9.95 | -14.91 | -14.22 | -9.44 | -7.44 | 24.97 | -14.22 | 2.84 | 19.94 | 16.45 | B-IO-S1 |
| a2.h11 | -1.52 | -1.37 | -1.22 | -1.20 | -1.04 | 0.60 | -1.22 | -0.74 | 0.13 | 0.21 | B |
| a2.h2 | -0.17 | -0.09 | -0.12 | 0.09 | -0.32 | 0.71 | -0.12 | 0.32 | 0.69 | 0.51 | OBJ-V |
| a2.h5 | -0.87 | -1.70 | -1.61 | -0.85 | -0.07 | 0.80 | -1.61 | -0.81 | 1.10 | 0.41 | E-M |
| a3.h3 | -0.51 | -0.68 | -0.71 | -0.44 | -0.61 | 0.95 | -0.71 | 0.08 | 0.40 | 0.35 | B-V |
| a3.h9 | -0.26 | -0.62 | -0.66 | -0.06 | -0.36 | 1.10 | -0.66 | 0.86 | 0.56 | 0.64 | B-V |
| a4.h0 | -0.87 | -0.87 | -1.04 | 0.13 | -1.26 | 1.41 | -1.04 | 0.31 | 2.02 | 0.88 | V |
| a4.h11 | -0.20 | -1.18 | -1.34 | -0.68 | -0.74 | 3.27 | -1.34 | 0.71 | 2.85 | 2.31 | OBJ |
| a4.h3 | -0.53 | -0.80 | -1.09 | 0.12 | -0.31 | 1.62 | -1.09 | 0.15 | 2.09 | 1.28 | OBJ |
| a5.h6 | -0.58 | -0.73 | -0.67 | -0.41 | -0.50 | 1.23 | -0.67 | -0.04 | 0.84 | 0.68 | B |
| a10.h0 | -0.56 | 26.75 | 7.03 | 2.09 | -0.40 | 0.09 | 7.03 | -0.85 | 0.11 | -0.21 | IO |
| a10.h10 | 0.83 | 21.84 | 5.49 | 0.63 | 0.04 | -8.69 | 5.49 | -1.48 | -6.38 | -6.35 | IO |
| a11.h3 | -0.38 | 7.43 | 5.20 | 4.21 | 0.24 | -1.10 | 5.20 | 0.20 | -0.65 | -0.80 | B-M |
| a6.h9 | -0.16 | 0.55 | 0.56 | -0.01 | -0.33 | -0.28 | 0.56 | -0.12 | -0.43 | -0.27 | B |
| a9.h6 | -1.55 | 21.43 | 5.85 | -0.36 | -1.76 | 3.52 | 5.85 | -0.28 | 3.33 | 2.43 | IO |
| a9.h9 | 0.59 | 46.30 | 3.95 | -0.31 | -0.36 | 1.58 | 3.95 | -0.70 | 0.22 | 0.67 | IO |
| a2.h4 | -0.22 | -0.19 | -0.19 | 0.15 | -0.10 | 0.41 | -0.19 | 0.43 | -0.27 | 0.11 | V |
| a5.h4 | -0.14 | -0.04 | -0.05 | 0.23 | -0.29 | 0.17 | -0.05 | 0.35 | 0.08 | 0.14 | B-V |
| a5.h5 | -0.04 | 0.13 | 0.30 | 0.38 | -0.31 | 0.27 | 0.30 | 1.02 | 0.28 | 0.07 | B |
| a5.h7 | -0.01 | 0.71 | 0.62 | 0.75 | -0.46 | 0.85 | 0.62 | 1.82 | -0.41 | 0.12 | B-V |
| a5.h8 | 0.64 | 0.46 | 0.56 | 0.78 | 0.53 | 0.80 | 0.56 | 1.14 | 0.76 | 1.04 | B |
| a6.h7 | 0.27 | 0.47 | 0.34 | 1.89 | 0.74 | -0.48 | 0.34 | 0.98 | -0.29 | 0.11 | B-OBJ |
| a7.h4 | 0.47 | 0.34 | 0.40 | 0.63 | 0.25 | 0.06 | 0.40 | 1.42 | 0.55 | 0.65 | B |
| a0.h3 | 0.87 | 2.25 | 2.25 | 1.52 | 1.18 | -3.60 | 2.25 | 0.42 | -2.82 | -2.46 | E-V |
| a0.h4 | 0.11 | 0.37 | 0.40 | 0.11 | -0.19 | -1.99 | 0.40 | -0.42 | -1.71 | -1.29 | E-V |
| a1.h11 | 0.80 | 2.79 | 2.76 | 1.68 | 1.03 | -6.74 | 2.76 | 0.25 | -5.67 | -4.81 | E |
| a1.h4 | 1.40 | 2.00 | 1.77 | 1.17 | 0.78 | -1.84 | 1.77 | 0.54 | -0.90 | -0.56 | B |
| a2.h9 | 0.60 | 0.72 | 0.48 | 0.75 | 0.61 | -0.24 | 0.48 | 0.85 | 0.70 | 0.48 | OBJ-V |
| a4.h4 | -0.28 | 0.34 | 0.44 | 0.53 | 0.27 | -1.31 | 0.44 | -0.43 | -1.12 | -0.69 | B |
| a4.h6 | 0.26 | 0.60 | 0.48 | 0.49 | 0.39 | -0.60 | 0.48 | -0.50 | 0.11 | 0.26 | B |
| a6.h4 | 0.30 | 0.78 | 0.87 | 0.91 | 0.34 | -0.65 | 0.87 | 0.48 | 0.15 | 0.10 | B |
| a7.h1 | 0.32 | 0.94 | 0.92 | 0.85 | 0.48 | -0.54 | 0.92 | 0.48 | -0.26 | -0.12 | B |
| a7.h3 | -0.36 | -0.24 | -0.49 | -1.41 | 0.41 | -1.73 | -0.49 | -3.62 | -1.20 | -1.19 | S2-V |
| a9.h4 | 0.61 | 1.91 | 2.17 | 0.16 | 0.66 | -2.12 | 2.17 | 1.07 | -0.70 | -0.63 | B |
| a9.h5 | -1.14 | -2.29 | -2.34 | -2.09 | -1.99 | -12.86 | -2.34 | -7.00 | -10.27 | -10.05 | B-S2 |
| | B | IO | S1 | OBJ | PLACE | M | S2 | V | E | T | Group |

Figure 25: BitFit Logit Clusters

Full Logit Lens Heatmap

| | B | IO | S1 | OBJ | PLACE | M | S2 | V | E | T | |
|---|---|---|---|---|---|---|---|---|---|---|---|
| FO | -96.93 | -69.60 | -81.19 | -91.40 | -91.72 | -88.09 | -81.19 | -90.49 | -87.23 | -86.56 | FO |
| a10.h2 | -3.89 | 4.66 | 6.88 | -3.54 | -2.27 | 4.85 | 6.88 | -0.58 | 4.02 | 3.30 | B-IO-S1 |
| a10.h6 | -0.91 | 5.13 | 2.58 | -0.19 | -3.59 | 2.65 | 2.58 | -0.06 | 3.13 | 2.66 | IO-S1 |
| a10.h9 | -0.56 | 1.57 | 1.56 | -0.04 | -0.19 | -0.34 | 1.56 | -0.88 | -0.04 | -0.23 | B |
| a11.h2 | -1.02 | 3.10 | 4.94 | -1.56 | -1.57 | 3.19 | 4.94 | -0.44 | 2.43 | 1.91 | B |
| a11.h3 | 0.53 | 10.98 | 7.80 | 4.20 | 0.56 | -2.58 | 7.80 | 0.57 | -2.25 | -1.96 | B-M |
| a6.h5 | -0.56 | 0.35 | 0.61 | -0.21 | 0.12 | -1.11 | 0.61 | -1.29 | -1.77 | -1.20 | B |
| a6.h9 | -0.17 | 0.35 | 0.41 | -0.00 | -0.31 | -0.35 | 0.41 | -0.30 | -0.37 | -0.31 | B |
| a8.h3 | 1.13 | 2.11 | 1.96 | -0.76 | -1.71 | 0.41 | 1.96 | 0.51 | 1.90 | 0.78 | IO-S1 |
| a9.h4 | 1.53 | 5.08 | 5.56 | 0.20 | 0.07 | -2.66 | 5.56 | 2.18 | -0.56 | -0.59 | B |
| a0.h8 | -1.09 | -1.83 | -1.94 | -2.24 | -1.80 | -2.57 | -1.94 | -0.82 | -1.66 | -1.00 | E-M |
| a1.h0 | 0.61 | 0.03 | -0.22 | 0.15 | -0.45 | 0.31 | -0.22 | 0.49 | 0.40 | 0.31 | V |
| a1.h2 | 0.75 | 0.59 | 0.43 | 0.67 | 0.67 | 1.00 | 0.43 | 0.66 | 0.72 | 0.76 | E |
| a2.h0 | 0.76 | 0.07 | -0.13 | -0.31 | -0.79 | 0.10 | -0.13 | -0.23 | 0.03 | -0.34 | B-E |
| a3.h4 | 0.34 | -0.67 | -0.65 | -0.65 | -0.32 | 0.24 | -0.65 | -0.60 | -0.09 | -0.25 | B |
| a4.h8 | 0.26 | 0.00 | 0.03 | 0.01 | -0.07 | -0.28 | 0.03 | 0.06 | -0.20 | -0.05 | B |
| a5.h9 | 0.94 | -1.28 | -1.28 | -1.28 | -0.45 | 0.47 | -1.28 | -1.43 | 0.23 | -0.03 | B |
| a0.h1 | -1.08 | -1.98 | -2.15 | -1.14 | -0.78 | -1.23 | -2.15 | -0.38 | -0.36 | -0.01 | E |
| a11.h10 | -0.77 | -9.05 | -8.34 | -1.12 | 0.92 | -0.78 | -8.34 | -0.48 | -1.04 | -0.50 | IO-S1 |
| a3.h5 | 0.09 | -0.18 | -0.13 | 0.32 | 0.10 | -0.30 | -0.13 | 0.25 | 0.20 | 0.16 | B |
| a3.h7 | 0.38 | 0.47 | 0.34 | 0.62 | 0.52 | 0.23 | 0.34 | 0.66 | 0.74 | 0.54 | OBJ |
| a4.h5 | -0.05 | -0.30 | -0.19 | -0.00 | -0.07 | -0.30 | -0.19 | -0.08 | 0.11 | -0.08 | B |
| a7.h9 | -0.53 | -2.92 | -6.19 | -0.62 | 0.10 | 3.07 | -6.19 | 2.29 | 2.75 | 3.14 | S2 |
| a8.h10 | -3.30 | -8.36 | -13.40 | -3.82 | -2.38 | -4.33 | -13.40 | -4.13 | -3.12 | -3.66 | S2 |
| a0.h10 | -0.29 | 0.14 | 0.19 | 0.07 | -0.65 | -0.97 | 0.19 | -0.04 | -1.26 | -0.95 | B-E |
| a0.h3 | 0.90 | 2.24 | 2.23 | 1.55 | 1.20 | -3.56 | 2.23 | 0.50 | -2.79 | -2.42 | E-V |
| a0.h4 | 0.03 | 0.31 | 0.32 | -0.06 | -0.23 | -2.42 | 0.32 | -0.56 | -2.03 | -1.54 | E-V |
| a1.h11 | 0.51 | 2.06 | 2.01 | 1.26 | 0.69 | -4.81 | 2.01 | 0.25 | -4.18 | -3.51 | E |
| a2.h10 | 2.27 | 3.26 | 3.17 | 2.52 | 2.72 | -2.87 | 3.17 | 1.83 | -1.67 | -0.62 | E-V |
| a2.h9 | 0.69 | 0.91 | 0.65 | 0.95 | 0.78 | -0.14 | 0.65 | 0.95 | 0.62 | 0.47 | OBJ |
| a3.h0 | -0.31 | 0.65 | 0.66 | 0.57 | 0.22 | -0.84 | 0.66 | 0.41 | -0.84 | -0.45 | B |
| a3.h10 | 0.54 | 1.89 | 1.82 | 1.27 | 1.32 | -2.64 | 1.82 | -0.21 | -2.38 | -1.60 | B-E |
| a4.h6 | 0.20 | 0.12 | 0.11 | 0.33 | 0.32 | -0.72 | 0.11 | -0.39 | 0.03 | 0.13 | B |
| a5.h0 | -0.51 | 0.29 | 0.23 | 0.30 | 0.40 | -0.73 | 0.23 | 0.10 | -0.34 | -0.10 | B |
| a6.h7 | 0.64 | 1.09 | 1.12 | 2.23 | 1.38 | -0.27 | 1.12 | 1.36 | 0.13 | 0.44 | B-OBJ |
| a7.h1 | 0.23 | 0.63 | 0.65 | 0.57 | 0.46 | -0.39 | 0.65 | 0.17 | -0.27 | -0.15 | B |
| a7.h3 | -0.28 | -0.48 | -0.76 | -1.17 | 0.23 | -0.98 | -0.76 | -2.72 | -1.21 | -0.84 | S2-V |
| a10.h0 | -0.06 | 32.32 | 8.08 | 1.75 | -0.75 | -1.76 | 8.08 | -1.51 | -1.45 | -1.38 | IO |
| a10.h10 | 0.52 | 25.66 | 4.24 | 0.70 | 0.48 | -5.91 | 4.24 | -0.41 | -4.64 | -4.55 | IO |
| a9.h6 | -0.24 | 26.11 | 5.35 | 1.13 | -1.00 | 2.86 | 5.35 | 0.71 | 2.71 | 1.97 | IO |
| a9.h9 | 0.38 | 47.03 | 4.44 | -0.03 | 0.06 | 1.62 | 4.44 | -0.59 | 0.01 | 0.57 | IO |
| a0.h0 | -0.94 | -1.12 | -1.09 | -0.76 | -0.89 | 1.04 | -1.09 | -0.42 | 0.42 | 0.39 | B-OBJ |
| a0.h2 | -0.78 | -1.58 | -1.62 | -1.00 | -1.09 | 1.64 | -1.62 | -0.32 | 1.50 | 0.83 | B-E-M |
| a0.h5 | -0.71 | -1.94 | -1.97 | -1.45 | -0.94 | 1.12 | -1.97 | -0.78 | 0.70 | 0.99 | E-T |
| a0.h6 | -1.29 | -1.34 | -1.32 | -0.85 | -0.79 | 1.85 | -1.32 | 0.21 | 0.61 | 0.46 | E |
| a0.h7 | -0.93 | -0.80 | -0.92 | -0.11 | -0.14 | 0.34 | -0.92 | -0.33 | 0.88 | 0.62 | E |
| a1.h3 | -2.08 | -2.64 | -2.57 | -2.08 | -2.09 | 1.88 | -2.57 | -1.34 | 1.19 | 1.19 | B |
| a1.h5 | -0.27 | -0.32 | -0.28 | -0.18 | 0.11 | 0.38 | -0.28 | 0.45 | 0.08 | 0.89 | E |
| a10.h1 | -1.83 | 3.48 | 0.07 | -1.42 | 0.15 | 8.50 | 0.07 | 2.04 | 5.93 | 5.47 | IO-S1 |
| a10.h3 | -6.94 | -8.74 | -8.18 | -6.32 | -4.90 | 14.39 | -8.18 | -0.39 | 12.23 | 9.80 | IO-S1 |
| a10.h7 | -4.40 | -5.98 | -5.31 | -4.66 | -3.20 | -1.94 | -5.31 | -4.83 | -1.94 | -2.04 | IO-S1 |
| a10.h8 | -3.65 | -4.96 | -5.00 | -3.18 | -2.14 | 10.04 | -5.00 | 0.56 | 7.18 | 7.49 | B-IO |
| a11.h1 | -5.65 | -7.46 | -7.05 | -5.51 | -2.74 | 13.89 | -7.05 | -0.32 | 11.63 | 9.65 | B |
| a11.h11 | -8.55 | -9.29 | -9.20 | -8.67 | -8.00 | 4.86 | -9.20 | -4.21 | 3.84 | 2.36 | B-M |
| a11.h5 | -6.72 | -9.46 | -9.21 | -5.78 | -5.07 | 18.15 | -9.21 | -2.22 | 14.92 | 11.61 | B-IO-S1 |
| a11.h6 | -2.06 | 0.12 | 0.78 | -2.50 | -2.81 | 4.57 | 0.78 | -0.04 | 3.27 | 2.57 | B-IO-S1 |
| a11.h9 | -8.47 | -15.07 | -15.39 | -8.63 | -6.97 | 21.39 | -15.39 | 1.11 | 17.10 | 14.03 | B-IO-S1 |
| a2.h2 | -0.27 | -0.22 | -0.26 | 0.03 | -0.44 | 0.80 | -0.26 | 0.22 | 0.64 | 0.50 | OBJ-V |
| a2.h5 | -0.85 | -1.85 | -1.75 | -0.96 | -0.08 | 1.01 | -1.75 | -0.87 | 1.17 | 0.40 | E-M-OBJ |
| a2.h6 | -0.79 | -0.69 | -0.76 | -0.24 | -0.00 | 0.32 | -0.76 | -0.62 | 0.33 | -0.19 | B-M |
| a2.h8 | -0.31 | 0.11 | -0.10 | 0.45 | -0.32 | 0.38 | -0.10 | -0.32 | 0.59 | 0.04 | V |
| a3.h2 | -0.59 | -0.92 | -0.86 | -0.44 | -0.99 | 0.62 | -0.86 | -1.00 | 0.97 | 0.48 | V |
| a3.h3 | -0.63 | -0.83 | -0.85 | -0.55 | -0.66 | 0.92 | -0.85 | -0.00 | 0.48 | 0.42 | B |
| a4.h0 | -0.80 | -0.83 | -0.97 | 0.13 | -1.20 | 1.61 | -0.97 | 0.39 | 1.92 | 0.93 | V |
| a4.h11 | -0.23 | -1.11 | -1.29 | -0.62 | -0.75 | 3.19 | -1.29 | 0.77 | 2.87 | 2.30 | OBJ |
| a4.h3 | -0.87 | -0.88 | -1.16 | 0.13 | -0.43 | 1.63 | -1.16 | 0.10 | 2.27 | 1.37 | OBJ |
| a5.h2 | -0.11 | -0.64 | -0.78 | -0.01 | -0.31 | 0.24 | -0.78 | -0.26 | 1.14 | 0.46 | B-OBJ |
| a5.h6 | -0.43 | -0.58 | -0.50 | -0.42 | -0.42 | 0.64 | -0.50 | -0.23 | 0.48 | 0.31 | B |
| a6.h0 | -0.36 | -1.35 | -1.49 | -0.56 | -1.12 | 2.00 | -1.49 | 0.53 | 2.13 | 1.59 | B-V |
| a6.h1 | -0.22 | 0.01 | 0.04 | 0.13 | -0.12 | 0.56 | 0.04 | 0.17 | 0.28 | 0.44 | B |
| a6.h8 | 0.62 | -0.05 | -0.13 | 0.15 | -0.35 | 0.43 | -0.13 | 0.32 | 1.37 | 0.91 | B-V |
| a7.h10 | -0.27 | -0.23 | -0.19 | -0.25 | -0.37 | 0.10 | -0.19 | 0.03 | 0.08 | 0.10 | B |
| a8.h5 | -1.45 | -0.60 | -0.67 | -0.09 | 0.12 | 2.70 | -0.67 | 1.55 | 3.31 | 2.43 | B-S2-V |
| a8.h6 | -2.58 | 0.52 | 1.11 | -4.24 | -5.77 | 10.67 | 1.11 | 0.71 | 8.73 | 7.86 | S2 |
| a8.h7 | -0.06 | -0.08 | -0.10 | 0.15 | -0.24 | 0.43 | -0.10 | 0.49 | 1.28 | 0.75 | B |
| a5.h10 | -1.23 | -1.04 | -1.23 | 2.15 | -0.90 | -0.14 | -1.23 | 0.67 | 0.93 | 0.30 | E-OBJ |
| a5.h5 | -0.29 | 0.03 | 0.23 | 0.32 | -0.44 | 0.22 | 0.23 | 1.17 | 0.43 | 0.03 | B |
| a5.h7 | -0.17 | 0.66 | 0.60 | 0.60 | -0.44 | 0.73 | 0.60 | 2.03 | -0.53 | 0.15 | B-V |
| a5.h8 | 0.51 | 0.41 | 0.53 | 0.85 | 0.53 | 0.64 | 0.53 | 1.02 | 0.66 | 0.87 | B |
| | B | IO | S1 | OBJ | PLACE | M | S2 | V | E | T | Group |

Figure 26: Full Logit Clusters

Figure 27: IA3 Logit Clusters

Figure 28: LoRA Logit Clusters

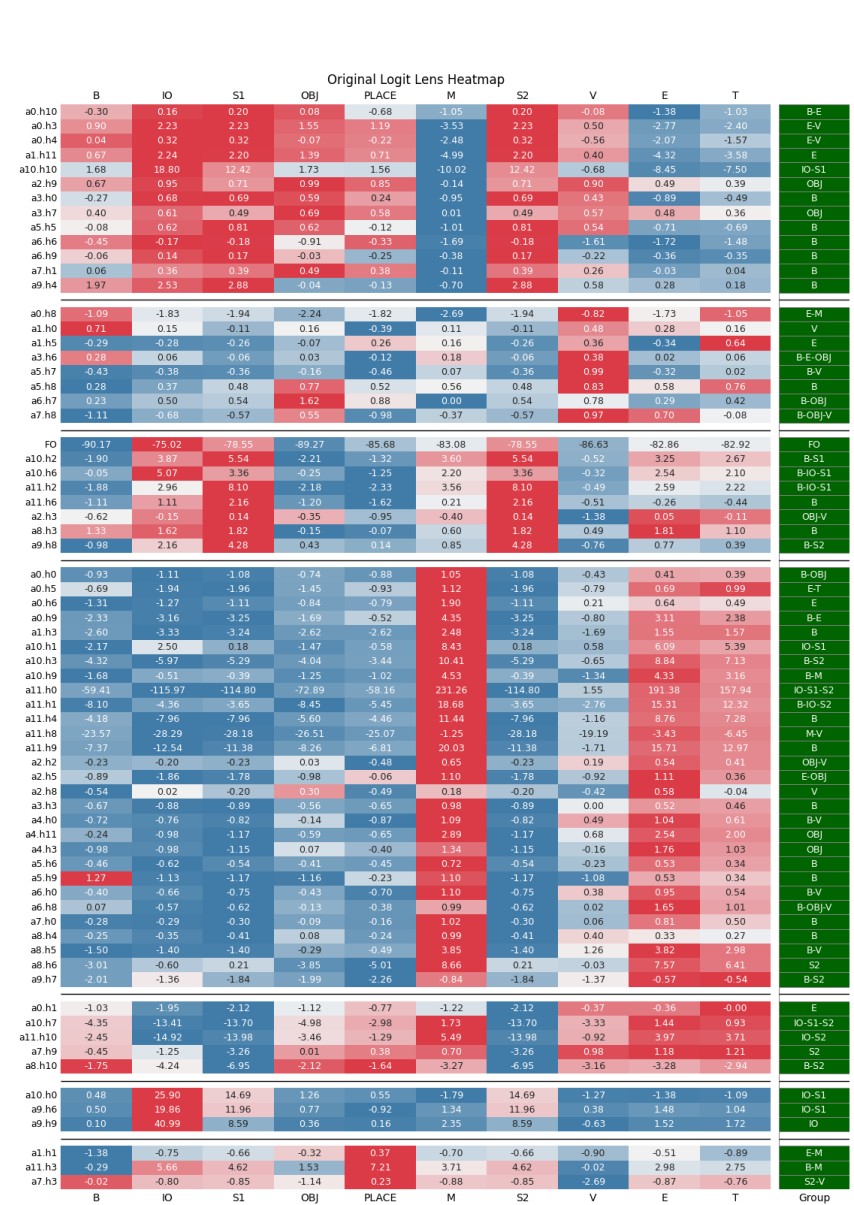

Figure 29: Original Logit Clusters

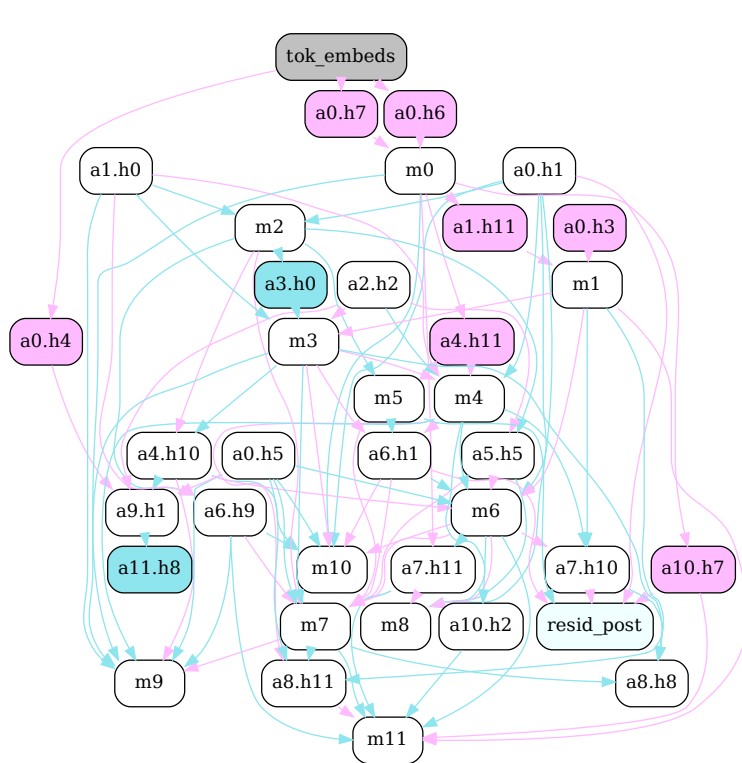

Figure 30: Circuit difference for GT task: LoRA vs Full. Blue indicates LoRA-specific components, red indicates Full-specific components, and white represents shared components.

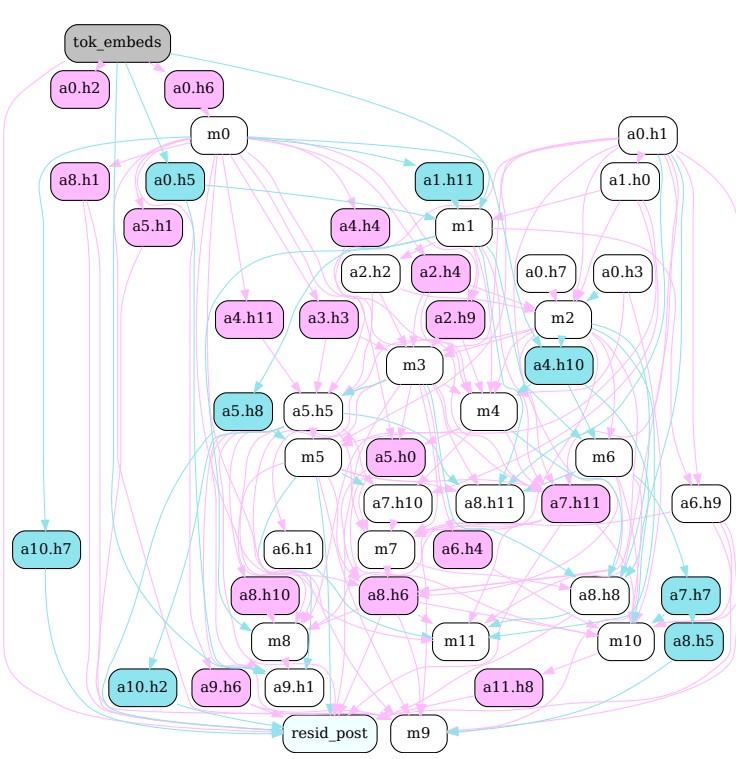

Figure 31: Circuit difference for GT task: Original vs AdaLoRA. Blue indicates Original-specific components, red indicates AdaLoRA-specific components, and white represents shared components.

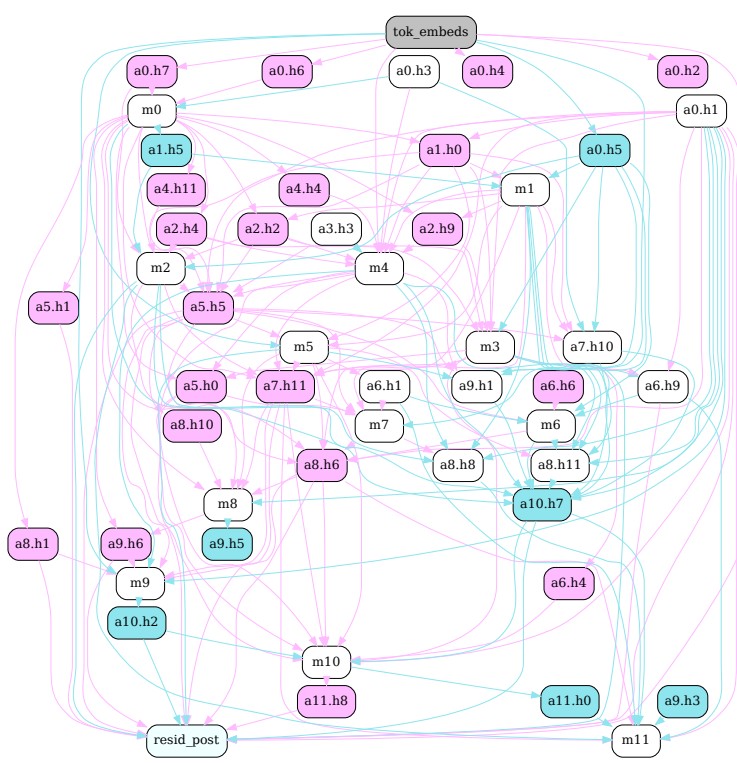

Figure 32: Circuit difference for GT task: Original vs BitFit. Blue indicates Original-specific components, red indicates BitFit-specific components, and white represents shared components.

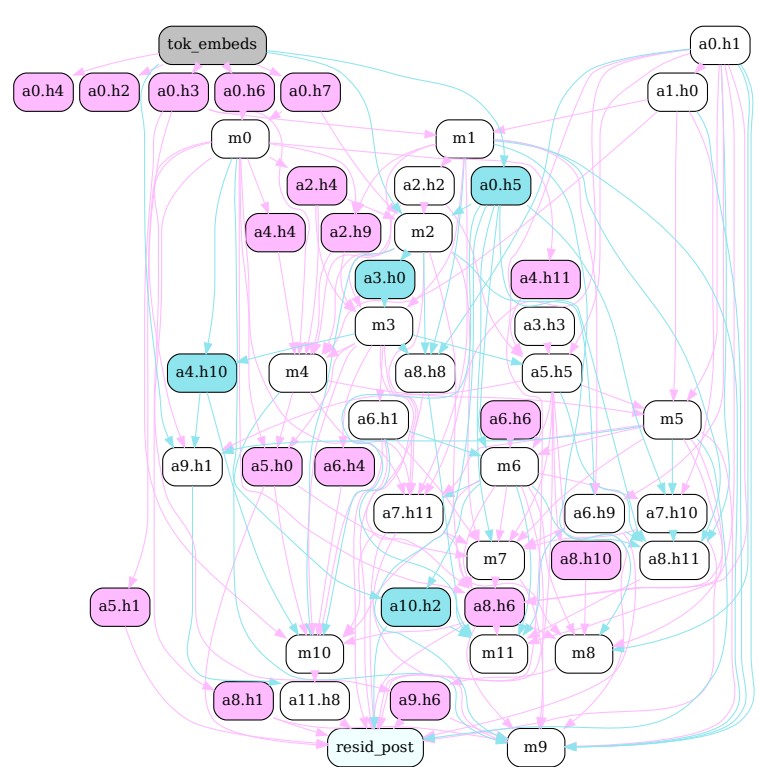

Figure 33: Circuit difference for GT task: Original vs Full. Blue indicates Original-specific components, red indicates Full-specific components, and white represents shared components.

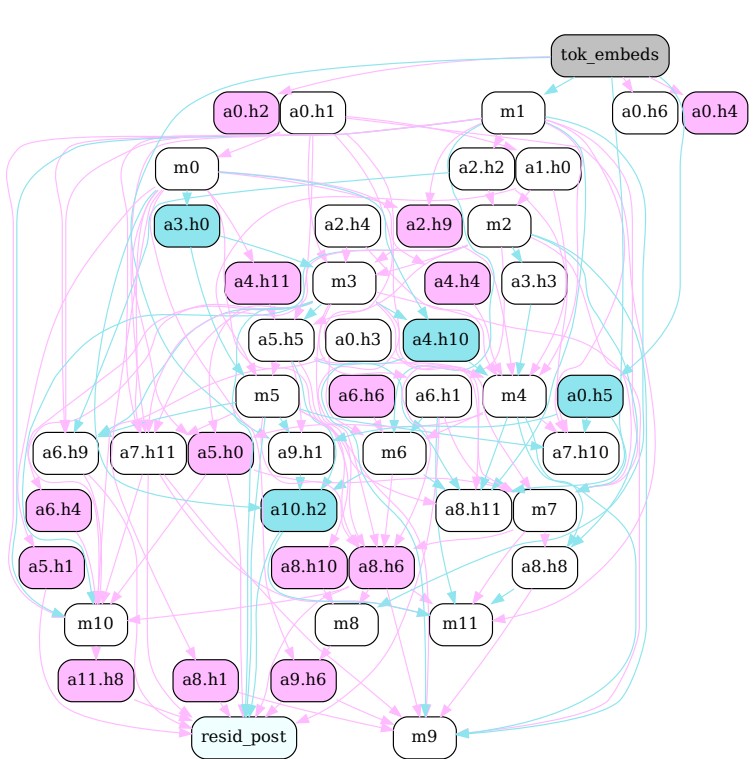

Figure 34: Circuit difference for GT task: Original vs IA3. Blue indicates Original-specific components, red indicates IA3-specific components, and white represents shared components.

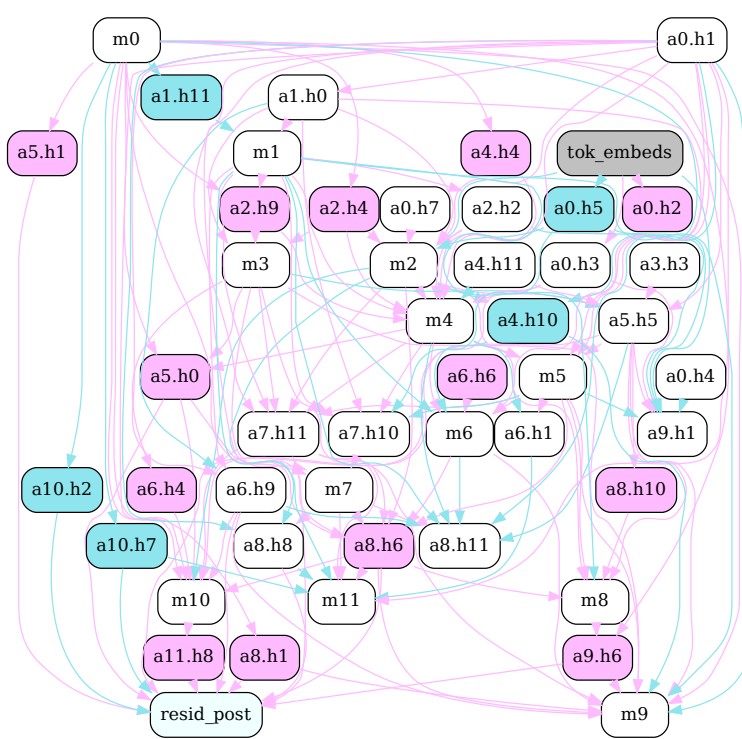

Figure 35: Circuit difference for GT task: Original vs LoRA. Blue indicates Original-specific components, red indicates LoRA-specific components, and white represents shared components.

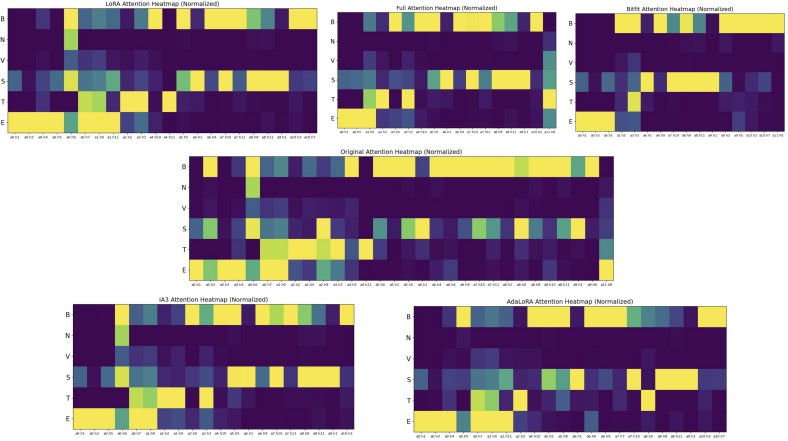

Figure 36: Attention grouping results for all finetuning methods circuit on GT.

