# OpenReview forum: "Mechanistic Insights: Circuit Transformations Across Input and Fine-Tuning Landscapes"
_ICLR.cc/2025/Conference — ICLR 2025 Conference Withdrawn Submission_

### Official Review · Reviewer_XHpQ · 2024-11-01

**Soundness:** 2
**Presentation:** 2
**Contribution:** 1
**Rating:** 1
**Confidence:** 4

**Summary:**

In their work, authors address the question of discovery and interpretability of model circuits responsible for problem solution, focusing on fine-tuning procedures like PEFT. Authors present a pipeline of edge pruning and circuit discovery starting from full model, leading to circuits identification for further analyses. Authors take two tasks, Indirect Object Identification (IOI) and Greater Than (GT), perform PEFT on GPT2 small base model and run the proposed circuit identification and analysis routines on control and fine-tuned models. Authors use performed analysis to compare various PEFT forms.

**Strengths:**

Question of interpretable circuit identification is an important direction, as it holds promise to reveal causes of model failures to solve certain problem types, also hinting how to fix model function and learning.

**Weaknesses:**

It is not clear from the presented work how the proposed circuit identification should help to better understand model function. Examples of interventions to improve model behavior based on conducted circuit analysis are missing. There is no comparison conducted to other SOTA methods for circuit identification and analysis. It is in general hard to understand from the paper what are the merits of proposed method and how it relates to already existing works.

**Questions:**

Is there any way to demonstrate how introduced methods can provide concrete invervention for model improvement which leads to better task performance on the 2 tasks that are studied? An example of such intervention based on derived circuits could give hints or proof of concept how presented method can be useful.

---

> ### Author Response · Authors · 2024-11-21
>
> Thank you for your comments. We are glad that you found our work addressing some important questions. We would like to make the following clarifications:
>
> >  It is not clear from the presented work how the proposed circuit identification should help to better understand model function.
>
> Our proposed method aims to streamline circuit analysis by automating key steps like logit clustering and attention grouping, allowing us to identify and characterize the functionalities of nodes more systematically. This is particularly important as research scales to larger models and more complex tasks, where manual inspection becomes infeasible. By identifying patterns and grouping nodes based on their functional roles, we provide insights into how specific components of the model contribute to task performance, shedding light on mechanisms like token emphasis.
>
> > Examples of interventions to improve model behavior based on conducted circuit analysis are missing.
>
> While this paper focuses on proposing the framework, future work could leverage the identified circuits to design targeted interventions. For instance: 1. Fine-tuning specific clusters of nodes to improve task performance or mitigate undesired behaviors. 2. Pruning or modifying circuits associated with incorrect outputs to enhance model robustness.
>
> However, we would like to emphasize that our work is aim to study the functionality of the nodes from the identified circuits. Interventions to improve model behavior is not the focus of our paper.
>
> > There is no comparison conducted to other SOTA methods for circuit identification and analysis.
>
> We apologize for any confusion. Our framework is compatible with any state-of-the-art (SOTA) method capable of generating circuits. As discussed in the Edge Pruning paper, which is one of the most recently released methods, it outperforms others such as ACDC in speed and attribution patching in accuracy and faithfulness. Therefore, we have chosen the most SOTA method for our framework.
>
>
> > It is in general hard to understand from the paper what are the merits of proposed method and how it relates to already existing works.
>
> Our method builds on prior works like IOI and GT by introducing automated steps that align with their core principles, such as attention analysis, while extending these ideas to larger and more complex circuits. Additionally, it could complement causal inference methods by providing a pre-analysis tool to narrow down focus areas for intervention or theoretical exploration.
>
> > Is there any way to demonstrate how introduced methods can provide concrete invervention for model improvement which leads to better task performance on the 2 tasks that are studied? An example of such intervention based on derived circuits could give hints or proof of concept how presented method can be useful.
>
> We are sorry for the confusion. As outlined in any of the circuit identification methods, the identified circuits are considered the part of the model that is responsible to perform a given task. So, finetuning this particular circuit or any some other reasonable interventions should improve the model performance. However, this is not the focus of our paper. Once again, we aim to propose a framework to study the functionalities of the nodes from identified circuits.

---

> > ### Comment · Reviewer_XHpQ · 2024-11-22
> >
> > I appreciate the time the authors took for the response. Given the assessment by other reviews and the rebuttal by the authors, I keep my score unchanged, as it remains hard for me to see what progress or new insights the work makes on the top of already existing work on mechanistic interpretability. I was also hoping for discussion on methods like Sparse Autoencoders in relation to circuit discovery and manipulation. I think works on mechanistic interpretability should in general develop own benchmarks containing canonical tasks of circuit identification and manipulation that would enable readers and community to compare proposed methods to already existing SOTA.

---

> > > ### Author Response · Authors · 2024-11-22
> > >
> > > Thank you for your response. We would like to reiterate that our work is not related to circuit discovery, circuit manipulation, or sparse autoencoders. Rather, it focuses on further analysis conducted after circuits have already been identified, an aspect that has been largely overlooked in the field. We kindly refer you to our paper and the introduction section for more details. We aim to develop a quantitative benchmark in future work.

---

### Official Review · Reviewer_hXPE · 2024-11-02

**Soundness:** 3
**Presentation:** 4
**Contribution:** 3
**Rating:** 8
**Confidence:** 3

**Summary:**

The authors build on previous circuit discovery methods to identify circuits for two commonly used template-specific tasks: IOI, and GT. They advance previous methods, which specialized on localization, by clustering relevant nodes into groups of similar functionality. For the IOI task, they test the generalization of found circuits to other subject domains, from human first names to animals (referred to as ablation). They further compare circuits found in the base model to circuits identified to a single model before and after task-specific fine-tuning.

**Strengths:**

In general, successful circuit discovery provides explanations for localization and functionality of circuit components. Existing circuit discovery methods mostly focus on localization (except for supervised methods like DAS). They are often template specific and sensitive to changes in the task. This paper addresses two key problems of existing work, which is relevant to the current state of the field:
1. Insight on node functionality
2. Variation of templates

- In my opinion, the main novelty is clustering attention heads by functionality. I think that further work on circuit discovery for templated tasks should use this method.
- This enabled a novel discovery of behavior shift during finetuning which could explain task-specific accuracy gains.
- The paper is well and clearly written, and provides useful descriptions of intuitions
- The figures generally support key points in the text well.

**Weaknesses:**

## Main critiques that can be addressed in this paper:
- In Table 4, I don’t understand what the “shared” column represents. My understanding is that it is the intersection of nodes (or edges?) in model A and B, but this doesn’t add up since often edges(A) + edges(B) < shared(A,B).
- Circuit selection could be quantitatively better described (probably in the Appendix due to space constraints?), How do the authors identify the ‘knee’ of KL divergence? From the current text, I cannot tell whether the authors simply visually picked circuits in that region or applied a more quantitative measure that combines faithfulness with sparsity.
- Section on logit clustering could contain more technical details. How did the authors aggregate over multiple IOI samples in a batch?

## Further critiques that mainly point towards future work.
- The scope of ablation studies is narrow. While it is a novel finding that animal IOI circuits are mostly a subset of human IOI circuits, the results do not provide enough evidence to judge whether the found circuits resemble “the complete IOI capability” of the model. Different templates, (like indicating IO with a verb in present progressive, eg. “The kind grandmother baked cookies for her grandchildren, delighting the grandchildren." This is not the best example, but varying the IOI template should be possible) would better address the general shortcoming of the field that circuits are too template specific.
- It’s sad that GPT-2 small is not able to do IOI for cities and colors and I appreciate that the authors included the performance results in the appendix. Studying more capable models should resolve this issue.
- I understand that studying GPT-2 small is a natural choice, since previous work used this model as well. The authors could make better use of this by highlighting parallels and differences to previously found circuits.
- Logit Lens can yield imprecise or misleading results due to residual stream drift. GPT-2 has tied embedding and LM-head weights, so residual stream drift should not be too much of a problem here. However, to be generally applicable, the authors approach of tying inThis method still depends on templated dataset and the manual investigation.
- Clustering focuses on attention heads, are MLPs treated in any novel way compared to previous methods? Are there reasons against applying Logit Clustering to MLPs?

**Questions:**

## Notes
- 307 “irrelevant” seems imprecise since the mentioned tokens (eg. M) provide useful grammatic informatic. Alternative: syntactically relevant or semantically irrelevant?
- Cite logit lens blogpost
- The authors mention “intended” functionalities of LLM attention heads? I’d be curious about a confidence assessment of how much the results determine the one specific intention in the authors’ opinion. I’m a bit skeptical, since same heads can fulfill different roles in different circuits
- Circuit diagrams (in most papers look) like a big blob, the field could improve on circuit visualizations
- In the appendix, the section headers don’t align with figures which appear on other pages further down. I can’t really make use of the appendix table of contents.

## Typos
065 what IS coming next
523 PATH patching

---

> ### Author Response · Authors · 2024-11-21
>
> Thank your for your comments! We are glad that you found our work interesting and having decent contribution to the field!
>
> > In Table 4, I don’t understand what the “shared” column represents. My understanding is that it is the intersection of nodes (or edges?) in model A and B, but this doesn’t add up since often edges(A) + edges(B) < shared(A,B).
>
> Sorry for the confusion! In the table, nodes(A) and nodes(B) are actually the number of unique nodes for circuits of model A and model B. And shared(A,B) are the number of shared nodes of A and B. The total of nodes from A would be nodes(A) + shared(A,B) for example. We will make this clearer in the revised draft.
>
> > Circuit selection could be quantitatively better described (probably in the Appendix due to space constraints?), How do the authors identify the ‘knee’ of KL divergence? From the current text, I cannot tell whether the authors simply visually picked circuits in that region or applied a more quantitative measure that combines faithfulness with sparsity.
>
> We identify the knee points by first fit a curve to the datapoints each represent a selected circuits then find the location where the second derivative is maximized. We then include some of the points around this estimated 'knee point'. However, we agree that there are more than one way to estimate such knee point. We will make this clear in the revised manuscript.
>
> > Section on logit clustering could contain more technical details. How did the authors aggregate over multiple IOI samples in a batch?
>
> We apologize for not including more technical details. We aggregate the samples in a batch by computing the average of logits over the samples.
>
> > The scope of ablation studies is narrow. While it is a novel finding that animal IOI circuits are mostly a subset of human IOI circuits, the results do not provide enough evidence to judge whether the found circuits resemble “the complete IOI capability” of the model. Different templates, (like indicating IO with a verb in present progressive, eg. “The kind grandmother baked cookies for her grandchildren, delighting the grandchildren." This is not the best example, but varying the IOI template should be possible) would better address the general shortcoming of the field that circuits are too template specific.
>
> We agree that the ablation studies in our work are still limited, and we acknowledge that more evidence is needed to draw definitive conclusions, such as whether the identified circuits fully represent “the complete IOI capability” of the model. Through our findings, we observed that animal IOI circuits may not resemble the original IOI circuits. Specifically, in ablated circuits, we noted that the cluster of nodes emphasizing IO or S tokens is often associated with tokens at the beginning of sentences. This behavior is uncommon in the original IOI circuits. We suspect that for animals or other ablated circuits, the model is more focused on identifying the location of IO and S tokens before performing reasoning tasks.
>
> > Clustering focuses on attention heads, are MLPs treated in any novel way compared to previous methods? Are there reasons against applying Logit Clustering to MLPs?
>
> We agree that there is no reason not to apply logit clustering to MLPs. We chose not to include an analysis of MLPs solely to better align with prior work.
>
> > 307 “irrelevant” seems imprecise since the mentioned tokens (eg. M) provide useful grammatic informatic. Alternative: syntactically relevant or semantically irrelevant?
> > Cite logit lens blogpost
>
> Thank you for pointing this out! We will modify the wordings with semantically irrelevant in the revised manuscript. And we will cite the logit lens blogpost.
>
> > In the appendix, the section headers don’t align with figures which appear on other pages further down. I can’t really make use of the appendix table of contents.
>
> We apologize for the mess in the appendix. We will clean those up in the revised manuscript.
>
> > 065 what IS coming next 523 PATH patching
>
> Thank you for catching this, we will correct the typos!

---

> ### Author Response · Authors · 2024-11-21
> **response cont.**
>
> > The authors mention “intended” functionalities of LLM attention heads? I’d be curious about a confidence assessment of how much the results determine the one specific intention in the authors’ opinion. I’m a bit skeptical, since same heads can fulfill different roles in different circuits
>
> We really appreciate your concern and share your skepticism regarding whether a head can consistently perform a single function across different tasks and circuits. This is exactly why we use "intended functionalities" rather than assuming fixed functionalities, aligning with the perspective that heads may adapt their roles depending on the task. To address confidence, we could evaluate the consistency of the "intended" functionalities by comparing nodes from circuits around the 'knee point' of a given task and model. This could involve measuring the logits associated with these nodes and estimating confidence intervals, providing a quantitative basis for assessing how consistently a head performs its intended role.

---

### Official Review · Reviewer_KcXA · 2024-11-03

**Soundness:** 2
**Presentation:** 2
**Contribution:** 2
**Rating:** 3
**Confidence:** 4

**Summary:**

The paper proposes a new framework for circuit analysis consisting of three stages: circuit selection and comparison, attention grouping, and logit clustering.

The stages are detailed as follows:

- **Circuit Selection and Comparison**: Circuits are evaluated using various metrics (e.g., KL divergence, Kendall's tau, etc) to determine their faithfulness to the original model at different sparsity levels. The optimal circuit is chosen using the elbow method, balancing sparsity and performance based on the selected metric.

- **Attention Grouping**: The prompt is divided into multiple parts, and attention patterns are computed for each part. Heads are considered to be attending to a specific component if their attention to it is above the mean of the top-k highest attention values. Groups are then formed based on whether the heads attend to the same group of tokens.

- **Logit Clustering**: Different heads are clustered based on their logit patterns. The idea is that heads with similar logit patterns likely up-weight or down-weight similar tokens.

The authors validate this method by applying it to analyze structural changes in circuits under various conditions. Specifically, they examine changes in circuit structure when prompts are modified (e.g., using animals instead of people) and when fine-tuning on specific tasks.

**Strengths:**

- The experimental section is well-chosen and provides several results that may interest the community. For example:
  1. Fine-tuning can lead to changes in circuit sizes, with variations depending on task complexity (sometimes larger, sometimes smaller).
  2. Fine-tuning can retain significant parts of circuits, sometimes strengthening their effects.
  3. The size of the circuits for IOI varies quite a bit with the ablated version with the animals version being much simpler, though still retaining a great deal of the components.

- The paper addresses a timely and relevant topic in circuit interpretability. The authors take a step toward automating circuit interpretation, an area often overshadowed by circuit discovery.

**Weaknesses:**

**Minor Weaknesses**
- The paper could benefit from clearer writing. It is sometimes difficult to follow. For example, on a first reading I had a hard time understanding what the authors proposed as part of their framework. Additionally, more mathematical descriptions would improve clarity. For instance, it’s unclear what exactly is being clustered in the logit clustering—whether it is the direct head output or the dot product between the unembedding matrix and logits. The paper right now has very little mathematical formulas and I think  adding more could aid the reader in understanding the method on top of the verbal descriptions.

- There are errors in the submission, such as missing content in Appendix A and Appendix D.

**Important Weaknesses**
- Some of the methodological contributions  of the framework seem small. For example, the circuit discovery and selection stage appears more like a simple check than a significant step in the framework, despite being principled.

- The attention grouping methodology has potential issues. Grouping heads across different layers can be misleading, as information may shift from one token to another one across layers. For example, a head might attend to token \( t \) in layer \( l \), but this information could shift to token \( t+1 \) in layer \( l+1 \), where another head attends to it. In this case, it would be incorrect to assign these two heads to different clusters as they are attending the same information (although at different positions). Hence, while attention patterns are useful, they are not be sufficient for rigorous analysis.

- Building on the previous point, while logit clustering and attention grouping are interesting, they appear more suited to exploratory analysis than to a formal interpretability framework. They function more as heuristics, which, while not inherently negative, differs from the authors' claims in the paper.

**Questions:**

- Could the authors provide more detail on what exactly is clustered in the logit clustering?
- How do the authors envision these methods integrating into the current interpretability workflow?

---

> ### Author Response · Authors · 2024-11-21
>
> Thank you for your comments, and we are glad that you found our paper addressing a timely topic in circuit interpretability. We would like to make some clarifications as below:
>
> > Some of the methodological contributions of the framework seem small. For example, the circuit discovery and selection stage appears more like a simple check than a significant step in the framework, despite being principled.
>
> However, the circuit discovery and selection stages are crucial steps in our framework. Our framework focuses on studying the functionalities of the nodes within those circuits. Therefore, ensuring that the circuits are faithful and accurate is essential, as it provides a solid foundation for the findings derived in the subsequent stages. Moreover, as automated circuit discovery becomes increasingly popular and feasible, having a group of circuits that are similar in metrics, as represented by each of the points in the circuit selection stage in Fig. 2, becomes inevitable. Our framework offers guidance on how to select some of these circuits for further analysis.
>
> > The attention grouping methodology has potential issues. Grouping heads across different layers can be misleading, as information may shift from one token to another one across layers. For example, a head might attend to token ( t ) in layer ( l ), but this information could shift to token ( t+1 ) in layer ( l+1 ), where another head attends to it. In this case, it would be incorrect to assign these two heads to different clusters as they are attending the same information (although at different positions). Hence, while attention patterns are useful, they are not be sufficient for rigorous analysis.
>
> We agree that relying solely on attention grouping is misleading. This is exactly why we also include logit clustering, with all our findings based on a combination of attention grouping and logit clustering. Our findings rely more heavily on logit clustering, with references drawn from attention grouping.
>
>  In fact, we chose attention grouping to align more closely with prior works such as IOI [1] and GT [2], which also partially rely on analyzing how each node attends to specific parts of the prompt to derive the functionality of the nodes. For example, in IOI, Wang et al. concluded: "Name Mover Heads are active at END, attend to previous names in the sentence, and copy the names they attend to. Due to the S-Inhibition Heads, they attend to the IO token over the S1 and S2 tokens."
>
> > Building on the previous point, while logit clustering and attention grouping are interesting, they appear more suited to exploratory analysis than to a formal interpretability framework. They function more as heuristics, which, while not inherently negative, differs from the authors' claims in the paper.
>
> We apologize for any confusion. Our work aims to help analyze the functionalities of the nodes in the circuits. Thus, we completely agree that this framework is more heuristic in nature. In fact, the framework is designed to reduce the burden of human inspection as research extends to more complex settings with significantly more and larger circuits to analyze. In reality, performing full causal inference on every single path of the circuits, comparing them with numerous others, and conducting a comprehensive analysis is infeasible. This is why we believe that a more heuristic framework like ours is necessary. Moreover, our framework does not hinder the implementation of more theoretical analyses, such as causal scrubbing. Instead, it can serve as a guide/hint for applying these more rigorous approaches.
>
> > Could the authors provide more detail on what exactly is clustered in the logit clustering?
>
> In logit clustering, we treat the logit output at each node for each token from the prompt as a vector and perform K-Means clustering on those vectors
>
> > How do the authors envision these methods integrating into the current interpretability workflow?
>
> As automated circuit discovery becomes more popular, experimental settings involving the analysis of more than ten or even hundreds of much larger circuits would become increasingly common. It is impractical to rely solely on human inspection with theoretical inference for each or any of these circuits. This is why our framework, which provides insights into the functionalities of nodes through grouping, is important. These insights can, of course, also serve as exploratory analyses and act as a starting point or guide for conducting more theoretical analyses.

---

> ### Author Response · Authors · 2024-11-21
> **response cont.**
>
> > The paper could benefit from clearer writing. It is sometimes difficult to follow. For example, on a first reading I had a hard time understanding what the authors proposed as part of their framework. Additionally, more mathematical descriptions would improve clarity. For instance, it’s unclear what exactly is being clustered in the logit clustering—whether it is the direct head output or the dot product between the unembedding matrix and logits. The paper right now has very little mathematical formulas and I think adding more could aid the reader in understanding the method on top of the verbal descriptions.
>
> We were mean to provide more intuitions than formulas when we were presenting the proposed frameworks. However, We apologize for the confusions. We will include more mathematical expressions into the revised manuscript.
>
> > There are errors in the submission, such as missing content in Appendix A and Appendix D.
>
> We apologize for this mistakes. We will correct those in the revised manuscripts.

---

> > ### Comment · Reviewer_KcXA · 2024-11-22
> >
> > I appreciate the time the authors took to comment on my response. However, after reading the other reviews and the rebuttal by the authors I retain my previous score as largely my assessment of the paper remains unchanged.

---

### Official Review · Reviewer_gWVV · 2024-11-04

**Soundness:** 2
**Presentation:** 3
**Contribution:** 1
**Rating:** 3
**Confidence:** 3

**Summary:**

The manuscript studies how transformer circuits discovered by the edge pruning algorithm (Bhaskar et al., 2024) change under ablated prompts and various types of fine-tuning methods (e.g., BitFit, LoRA, full). The manuscript proposes a framework to study the functionalities of various discovered circuits. Empirical results are observed after prompt ablation and fine-tuning on GPT-2 small, where statistically significant variations in structures of discovered circuits.

**Strengths:**

- Transformer interpretability is a novel field with very interesting results and of great relevance to the audience of this venue
- Very detailed analysis, both quantitative and qualitative, explaining the framework

**Weaknesses:**

- The paper reads like an exposition on a series of analyses performed on a particular transformer on a set of prompts. The main result only showed that the proposed "perturbation" (both parameters and input data) has caused statistically significant variation in the discovered circuit from Bhaskar et al. I failed to see the significance of such variation, nor did the author do a good job explaining them: intuitively, such variation is either naturally straightforward, or an artifact of edge pruning, and not inherently offering insights on why they are occurring, and how that may lead to either better designs of the network architecture, data pipeline, or training algorithm. Granted that the authors have spent painful details qualitatively measuring logit clusters and rendering individual circuits from their experience, but it is very difficult to reason about the significance of these particular observations from the experiments.
- Hence, the claim that this is a novel framework that accelerates circuit discovery and analysis is very weak since the analysis appears to be superficial, and the framework appears to be a set of procedures that the authors elected to apply.
- The work also builds heavily on edge pruning - it may better be characterized as a technical report, as an application of Bhaskar et al., rather than a novel contribution.
- Finally, the experiments are only performed on a small transformer. It would be more convincing for the authors to provide additional experiments that may suggest that the proposed "framework" can indeed scale to larger networks.

**Questions:**

What is the main takeaway from applying the proposed framework on various versions of a transformer / a transformer on a set of ablated prompts? Beyond that, "the discovered circuits are different" and that "particular circuits appear to form different attention groups and logit clusters."

---

> ### Author Response · Authors · 2024-11-21
>
> Thank you for your comments. We are glad that you found our analysis detailed, both qualitatively and quantitatively.
>
> We apologize if our extensive analysis of numerous circuits was overwhelming. We understand that presenting analysis of six different circuits for each task type, along with their ablated versions, might have caused confusion. We would like to make some clarifications:
>
> > The paper reads like an exposition on a series of analyses performed on a particular transformer on a set of prompts. The main result only showed that the proposed "perturbation" (both parameters and input data) has caused statistically significant variation in the discovered circuit from Bhaskar et al. I failed to see the significance of such variation, nor did the author do a good job explaining them: intuitively, such variation is either naturally straightforward, or an artifact of edge pruning, and not inherently offering insights on why they are occurring, and how that may lead to either better designs of the network architecture, data pipeline, or training algorithm. Granted that the authors have spent painful details qualitatively measuring logit clusters and rendering individual circuits from their experience, but it is very difficult to reason about the significance of these particular observations from the experiments.
>
> > What is the main takeaway from applying the proposed framework on various versions of a transformer / a transformer on a set of ablated prompts? Beyond that, "the discovered circuits are different" and that "particular circuits appear to form different attention groups and logit clusters."
>
> First of all, Bhaskar et al. introduce a method for automatically identifying the circuits for different tasks, which is called edge pruning. We use this method to identify circuits for various tasks and perform analyses between those circuits. The circuits reported in this paper are all derived by implementing edge pruning, as introduced by Bhaskar et al.
>
> Our main finding goes beyond demonstrating that the circuits differ; it reveals that through ablated prompts or different fine-tuning methods, models approach similar tasks in distinct ways. For example, as discussed in Section 3.1, we observed that replacing human names with animals leads the model to identify IO and S tokens by first scanning the beginning of the sentence. This behavior contrasts with the original IOI task, where the model easily identifies IO and S token locations without prioritizing the start of the sentence. This observation also explains why models without fine-tuning struggle to maintain decent performance when IO and S tokens are replaced with other objects, such as colors or bridges.
>
> As discussed in Section 3.2, we found that fine-tuning, while preserving the overall structure of functionalities, statistically significantly introduces more circuits, such as nodes that up-weight the logits of IO (e.g., name movers) for the IOI task. It also results in much smaller circuits than the original ones when performing GT tasks, which are simpler than IOI, as shown in Table 1. Furthermore, as demonstrated in Fig. 4, fine-tuning reduces the complexity of the circuits identified for ablated prompts. This suggests that ablated prompts may transform the task into simpler ones, such as basic IO identification, whereas the original IOI tasks require more complicated reasoning.
>
> > Hence, the claim that this is a novel framework that accelerates circuit discovery and analysis is very weak since the analysis appears to be superficial, and the framework appears to be a set of procedures that the authors elected to apply.
>
> We apologize if this was not made clear in our paper. Our work does not aim to accelerate circuit discovery; instead, it focuses on automating certain steps in circuit analysis, particularly in identifying and understanding differences between circuits. This is crucial as research extends to more complex settings, where it becomes inevitable to analyze multiple circuits, unlike earlier works such as IOI [2] and GT [3]. These prior studies relied entirely on human inspection to identify and classify node functionalities. However, as the number of circuits increases, manual inspection becomes impractical, as we discussed in our motivation. To address this challenge, we proposed a framework to streamline and accelerate this step in the analysis process.
>
> And we respectively disagree that our analysis is superficial. As we clarified above, it is not a study showing A is smaller than B. As you also mentioned in the review, it requires tons of quantitative and qualitative analysis.

---

> ### Author Response · Authors · 2024-11-21
> **Response Cont.**
>
> > The work also builds heavily on edge pruning - it may better be characterized as a technical report, as an application of Bhaskar et al., rather than a novel contribution.
>
> We apologize for the confusion. We did not claim that the novelty lies in the integration of edge pruning. Instead, the novelty of our work stems from the logits clustering, attention grouping strategies, and the in-depth analysis of the circuits as noted by reviewer **hXPE**.
>
> > Finally, the experiments are only performed on a small transformer. It would be more convincing for the authors to provide additional experiments that may suggest that the proposed "framework" can indeed scale to larger networks.
>
> As demonstrated through edge pruning [1], circuits can be discovered even in billion-parameter models. Having a larger circuit does not hinder the implementation of our framework; its effectiveness depends more on whether the circuit discovery methods can scale appropriately.
>
> [1] Kevin Wang, Alexandre Variengien, Arthur Conmy, Buck Shlegeris, and Jacob Steinhardt. Interpretability in the wild: a circuit for indirect object identification in gpt-2 small. arXiv preprint
> arXiv:2211.00593, 2022.
>
> [2] Michael Hanna, Ollie Liu, and Alexandre Variengien. How does gpt-2 compute greater-than?: Interpreting mathematical abilities in a pre-trained language model. Advances in Neural Information
> Processing Systems, 36, 2024.
>
> [3] Adithya Bhaskar, Alexander Wettig, Dan Friedman, and Danqi Chen. Finding transformer circuits
> with edge pruning. arXiv preprint arXiv:2406.16778, 2024.

---

> > ### Comment · Reviewer_gWVV · 2024-11-26
> >
> > Thank you for your responses. I've decided to maintain my score. This is a work of good quality and great details, but of limited novelty and interest to the audience of this venue.

---

### Note · Authors · 2024-11-26

I have read and agree with the venue's withdrawal policy on behalf of myself and my co-authors.